# Minimum Displacement in Existing Moment (MDEM)- A new supervised learning algorithm by incrementally constructing the moments of the underlying classes

**Ahmed Mehedi Nizam** ⬤ *

The Central Bank of Bangladesh, Motijheel, Dhaka, Bangladesh

* ahmed.mehedi.nizam@gmail.com

## Abstract

We introduce a supervised learning method that classifies each test point by selecting the class for which its inclusion causes minimum displacement of the class's existing n-th central moment. After each such inclusion, the n-th central moment of the corresponding class is updated by some incremental calculations in constant time, i.e., each class evolves gradually and changes its definition incrementally after the inclusion of every new data point. We then use k-fold and stratified k-fold cross validation techniques to compare the performance of our proposed model with various state of the art supervised learning algorithms including Neural Network (NN), Support Vector Machine (SVM), Random Forest (RF), K-Nearest Neighbor (KNN) and Logistic Regression (LR) using Pima Indian Diabetes (PID) dataset and Wisconsin breast cancer dataset, which are popular datasets in machine learning research. Our analyses suggest the performances of different MDEM algorithms as proposed here involving different order of moments vary within the range of $[83.19\% - 95.82\%]$ of the best algorithm under consideration in k-fold and stratified k-fold cross validation techniques for PID dataset. Moreover, for Wisconsin breast cancer dataset, different variants of MDEM algorithms have achieved accuracy scores in the range of $[88.85\% - 96.41\%]$ of the best algorithm. Finally, we compare the results produced by different algorithms by constructing the corresponding confusion matrices.

## 1 Introduction and scope of the current study

To date, there are numerous supervised machine learning algorithms each having its own strength and weakness. For example, K-Nearest Neighbor (KNN), one of the earliest and perhaps most straightforward algorithm for supervised learning has the ability to train itself in constant time, i.e., as soon as a labelled input is provided, the model can learn it instantly without further processing. However, it has a testing complexity of $O(nd)$, where $n$ is the total number of training data, $d$ is the dimensionality of the feature space. This is a rather daunting task, specifically when we have a

**Data availability statement:** All data used in this study are publicly available and can be accessed from the Pima Indians Diabetes Database (https://www.kaggle.com/datasets/uciml/pima-indians-diabetes-database) and the Breast Cancer Wisconsin (Diagnostic) Data Set (https://www.kaggle.com/datasets/uciml/breast-cancer-wisconsin-data).

**Funding:** The author(s) received no specific funding for this work.

**Competing interests:** The authors have declared that no competing interests exist.

large set of training data, as we need to calculate the distance between the new point and all other previously classified points[1]. Using techniques like KD Tree and Ball Tree, average running time in the testing phase can be improved up to $O(\log n)$ at the expense of a costlier time complexity in the training phase [2,3]. However, the worst-case time complexity is still $O(n)$ [2,3].

Apart from the KNN, another most studied supervised learning algorithm is Support Vector Machine (SVM) that aims to construct maximum margin hyperplanes amongst the training data points [4]. The time complexity of SVM method depends, among other things, upon the algorithm used for optimization (e.g., quadratic programming or gradient-based methods), dimensionality of the data, the type of the kernel function used, number of support vectors, and the size of the training dataset. Worst-case time complexity in training phase of linear and non-linear SVMs are found to be $O((n^2 \times d))$ and $O(n^2 d)$–$O(n^3)$ respectively, where $n$ is the size of the training sample and $d$ is the dimensionality of the feature space, although the time complexity can be improved upon using various stochastic optimization technique [5,6]. Time complexities of the testing phase of SVMs are found to be $O(d)$ for linear kernel and $O(s \times d)$ for non-linear kernel, where $s$ is the number of support vectors.

Perhaps one of the most straightforward algorithm for supervised learning is the Nearest Centroid Classifier (NCC) that attempts to estimate the centroid of each class in the training phase, which can be done in $O(n \times d)$ time, where $n$ is the number of training samples and $d$ is the dimensionality of the feature space [7]. Moreover, in NCC, every new instance can be classified in $O(c \times d)$ time, where $c$ is the number of centroids, as it involves computing the Euclidean distance between each centroid and the new sample under consideration [8].

Another algorithm that is frequently used in classification of labelled data is Random Forest (RF) that works by constructing multiple decision trees in the training phase, where each tree is trained with a subset of the total data [9]. To predict the final output of the RF in testing phase, majority voting technique is used to combine the results of multiple decision trees. The time complexity of RF depends upon the number of trees in the forest ($t$), sample size ($n$), dimensionality of the feature space ($d$) and tree height ($h$) among other things. Training time complexity of RF is found to be $O(t \times n \times d \times h)$, while the testing complexity is $O(t \times h)$ per sample.

Another important algorithm for classification is Logistic Regression (LR), which is used primarily for binary classification, although the algorithm can be easily adapted to handle multiclass classification problems. Training phase time complexity of the Logistic Regression (LR) depends, among others, on number of training samples, number of iterations and dimensionality of the features space, where the number of iterations depends further on the choice of the algorithm used (stochastic, batch gradient descent or, alike) [10]. In a nutshell, the total time complexity of the Logistic Regression (LR) in training phase can be summarized as $O(E \times n \times d)$, where $E$ is the number of iterations, $n$ is the size of the training sample and $d$ is the dimensionality of the input space. On the other hand, the testing time complexity of LR per sample is $O(d)$ as it simply involves computing the dot product of the weight ($w$) and feature vector ($x$) [10,11].

However, perhaps one of the most popular and widely used supervised learning algorithms is the Neural Network (NN), which is inspired from the networks of biological neurons that comprise human brain and is presently used extensively in image and video processing, natural language processing, healthcare, autonomous vehicle routing, finance, robotics, gaming and entertainment, marketing and customer service, anomaly detection etc. Performance of a Neural Network (NN) depends upon the number of hidden layers, number of neurons per layer, number of epochs, input size, input dimensions etc. If there are $L$ hidden layers each having $M$ neurons, then the training time complexity of the Neural Network (NN) can be summarized as $O(L \times M \times E \times n \times d)$, where $E$ is the number of epochs/iterations, $n$ is the sample size and $d$ is the dimensionality of the input space, while the testing time complexity per sample of the said NN is $O(L \times M \times d)$ [12,13]. Choices of the number of hidden layers $L$, number of neurons per layer $M$ and number of epochs $E$ are somewhat arbitrary, i.e., we can choose any value for $L, M$ and $E$ from a seemingly infinite range.

In fact, all of the above algorithms apart from KNN have one or more arbitrary parameters to be set, e.g., number of iterations, number of trees, choice of optimization algorithm, choice of kernels, number of hidden layers, number of nodes in each hidden layer, choice of activation function etc. Although KNN has a deterministic training and testing complexity, which can be anticipated beforehand, its testing time complexity is linear on training space, which is a very time-consuming process and renders KNN effectively ineffective in case of large training data. Here, we propose a new supervised learning algorithm that has a deterministic running time and can learn in $O(nd)$ time and classify new inputs in $O(kd)$ times, where $n$ is the number of inputs, $d$ is the dimensionality of input space and $k$ is the number of classes under consideration. For a specific problem, the dimensionality of input space $d$ and the number of classes $k$ are fixed. Thus, unlike KNN, the training phase time complexity of our proposed algorithm is linear on the number of inputs and the testing time complexity per sample is constant. So, whenever we need a light-weight deterministic algorithm like KNN that, unlike KNN, can effectively classify new instances in constant time, we can use our proposed algorithm, which does not involve solving a complex quadratic programming problem (as like SVM) or operations that require matrix multiplication (for NN) or the alike.

In the training phase, our proposed algorithm resorts to find the $n$-th moment (raw or central) of each attribute of every class. At testing phase, the algorithm temporarily includes the new input into each of the $k$ classes and computes the new, temporary $n$-th moment for each attribute of each class after a temporary inclusion of the new data point into every possible class. The new input will then be finally classified into the class for which such inclusion causes minimum displacement in the existing $n$-th moment of the underlying class attributes. Once the new input is classified, the $n$-th moment of the attributes of the respective class is updated to reflect the change, while the moments of all other classes are left unchanged. Thus, apart from classifying new input in constant time, our algorithm also evolves incrementally after inclusion of every new data point, which makes the model dynamic in nature.

The rest of the article is organized as follows: Sect 2 provides the definitions of raw and central moments as used in our analysis, Sect 3 describes the new algorithm for supervised learning based upon the Minimum Displacement in Existing Moment (MDEM) technique as improvised here, Sect 4 discusses the time complexities of the proposed algorithm, Sect 5 provides the theoretical foundation of the proposed MDEM algorithms, Sect 6 derives a sufficient condition for optimality under minimum perturbation criteria of MDEM algorithms, Sect 7 presents the methodology used for empirical analysis, Sect 8 describes and elaborates the data, Sect 9 presents various preprocessing techniques used for data cleansing, Sect 10 discusses the empirical results and compares the performance of our proposed algorithms to that of various state of the art supervised learning techniques including K-Nearest Neighbor (KNN), Support Vector Machine (SVM), Random Forest (RF), Logistic Regression (LR) and Neural Network (NN), and finally, Sect 11 concludes the article.

## 2 Raw and central moments

### 2.1 Raw moment

$n$-th Raw moment $\mu'_n$ of a random variable $X$ is defined as the expected value of the $n$-th power of $X$, i.e.,

$$\mu'_n = E[X^n]$$

When $n = 1$, the first raw moment $\mu'_1$ of $X$ is the mean of the random variable $X$.

### 2.2 Central moment

$n$-th central moment $\mu_n$ of a random variable $X$ is defined as the expected value of the $n$-th power of the deviation of the random variable $X$ from its mean. Mathematically,

$$\mu_n = E[(X - \mu)^n]$$

When $n = 1$, the first central moment of the random variable $X$ about its mean is zero and when $n = 2$, the second central moment is the variance of the random variable $X$. For any arbitrary $n$, if we extend the expression of $E[(X - \mu)^n]$ using binomial theorem, we get the following expression:

$$E[(X - \mu)^n] = E\left[\sum_{k=1}^{n} \binom{n}{k} X^{n-k} (-\mu)^k\right] \tag{1}$$

Using the linearity property of the expectation operator, we can simplify Eq 1 for $n = 2$ as follows:

$$E[(X - \mu)^2] = E[X^2] - \mu^2 \tag{2}$$

For $n = 3$, Eq 1 can be simplified into the following form to get the 3rd central moment about the mean:

$$E[(X - \mu)^3] = E[X^3] - 3E[X^2]\mu + 2\mu^3 \tag{3}$$

And for $n = 4$, Eq 1 can be simplified as below to get the 4th central moment about the mean:

$$E[(X - \mu)^4] = E[X^4] - 4\mu E[X^3] + 6E[X^2]\mu^2 - 3\mu^4 \tag{4}$$

We use Eqs 2, 3 and 4 to calculate 2nd, 3rd and 4th central moments respectively as part of our current proposition.

As we have discussed above, the first raw moment and second central moment of a distribution are its mean and variance respectively. Mean and variance are two important statistics of the distribution that captures important information about the distribution. Moreover, in this regard, we may recall that the third central moment of a distribution is popularly known as the (non-normalized) skewness, which represents how symmetric the distribution is with respect to its mean. If the distribution has zero skewness, then the data points are evenly distributed on both sides of the mean. Besides, positive skewness (or right skew) indicates a longer tail on the right side, while negative skewness (or left skew) indicates a longer tail on the left side.

On the other hand, the fourth central moment of a probability distribution is known as its kurtosis in non-normalized form. Kurtosis represents the heaviness of the two tails as well as the sharpness of the peak of a probability distribution as compared to a normal distribution. If a distribution has higher kurtosis, i.e., heavier tail and sharper peak than a normal

distribution, then it means, it has a higher concentration of outliers than a normal distribution and is known as Leptokurtic distribution. On the other, if a distribution has low kurtosis, i.e., lighter tails and flatter peaks than a normal distribution, then it is supposed to have fewer outliers and less variability from its mean.

Four moments mentioned above, namely, mean, variance, skewness and kurtosis are four very important sample statistics and they say a lot about the underlying distribution. So, we consider all of these attributes in our current study, although we can equivalently consider any other higher order moments. In our approach, every new point is classified into the class that is least perturbed in terms of mean, variance, skewness, kurtosis as a result of such inclusion.

## 3 Proposed supervised learning algorithm based upon Minimum Displacement in Existing Moment (MDEM)

We begin our analysis by sketching the algorithm for the 1st raw moment, i.e., mean. In this step, we intuitively describe the main idea behind the current discourse and then in subsequent steps, we enhance the reasoning to account for higher order central moments of the classes under construction.

### 3.1 MDEM in mean

Let us assume that we are attempting to solve a multiclass classification problem involving $M$ different classes. The classes are enumerated from 1 to $M$. Let us also assume that each of the training instances has $N$ number of numerical attributes. Based upon the value of these $N$ attributes, each training instance is classified into one of the $M$ possible classes. To begin with, we scan each of the training inputs one at a time and incrementally calculate the sum of the attributes of the respective class. Sum of all of the $k$-th attributes of the $j$-th class is stored into $Sum[j][k]$. This is done through line [10–14] of Algorithm 1. As soon as a new training instance is found to belong to class $j$, the $count[j]$ is increased by one. After we are done with scanning of the training rows, we calculate the mean of each attribute of every class. This is done simply by dividing the $sum[i][j]$, $\forall i, 1 \leq i \leq M, \forall j, 1 \leq j \leq N$ by $count[i]$, $\forall i, 1 \leq I \leq M$. This is done in line [15–17] of Algorithm 1. This marks the end of the training phase of our algorithm.

At the end of the training phase, we have an array of $Mean[][]$, which is a 2D array containing the mean of $j$-th attribute of $i$-th class at $Mean[i][j]$. We also have another 2D array, namely, $Sum[][]$ containing the sum of all of the $j$-th attributes belonging to $i$-th class at $Sum[i][j]$. In the testing phase, we incrementally enhance the $Sum[i][j], \forall j, 1 \leq j \leq N$ as soon as a new instance is classified into class $i$ by our proposed algorithm. Thus, after every increment, our algorithm evolves to accommodate the new changes. At the very beginning of the testing step, we scan a new test row and temporarily include it into every possible class. This will allow us to calculate the new temporary mean of every attribute of each class after such *pseudo inclusion*. This is done in line [2–5] of Algorithm 2. Next, we find the Euclidean distance between the existing mean and new temporary mean for each class. This is to be noted in this regard that both $Mean[k]$ and $tMean[k]$, $\forall k, 1 \leq k \leq M$ are $N$-dimensional vectors of the attributes. This is done in line [6–8] of Algorithm 2.

Euclidean distances thus calculated need to be multiplied by the cardinality of the respective class lest the class with high cardinality *eat up every new point due to its gravitational pull*. This is diagrammatically presented in Fig 1. In Fig 1, we have two classes, namely, Class-1 and Class-2. Class-1 already has 46 members in it, while Class-2 only has 2 members. As soon as we have a new point A to be classified, we notice that the inclusion of point A causes very little displacement in the existing mean of Class-1 as compared to Class-2. To be precise, inclusion of point A into Class-1 causes its mean (Class-1's mean) to be shifted from point 1 to point 2. However, if point A is instead classified into Class-2, then its mean (Class-2's mean) is shifted from point 3 to point 4. As evident from Fig 1, distance between [1,2] is quite small as compared to that of [3,4]. So, apparently at this point, we may consider the new point A to be classified into Class-1. However, as we can visually comprehend from Fig 1, point A is supposed to be classified into Class-2. To resolve the issue, we multiply the Euclidean distance calculated as above by the cardinality of the respective class. This modification of weighting the Euclidean distance by the class cardinality is done in line 8 of Algorithm 2.

**Algorithm 1.  Pseudocode for MDEM in mean: training phase.**

**Data:** `TrainingData, TestingData`
**Result:** `Mean[M][N]`
/* Initialize data structures                                                                  */
1 **for** $i \leftarrow 1$ **to** $M$ **do**
2 **for** $j \leftarrow 1$ **to** $N$ **do**
3 $Mean[i][j] \leftarrow 0$ // $M$ is number of classes, $N$ is number of attributes
4 $tMean[i][j] \leftarrow 0$ // $tMean$ is temporary mean
5 $Sum[i][j] \leftarrow 0$;
6 $tSum[i][j] \leftarrow 0$;
7 $count[i] \leftarrow 0$;
8 $Train[TrainRows][N] \leftarrow$ `TrainingData`;
9 $Test[TestRows][N] \leftarrow$ `TestingData`;
/* Beginning of training phase                                                                  */
10 **for** $i \leftarrow 1$ **to** $TrainRows$ **do**
11 **if** $i$-th training instance belongs to class $j$ **then**
12 **for** $k \leftarrow 1$ **to** $N$ **do**
13 $Sum[j][k] \leftarrow Sum[j][k] + Train[i][k]$;
14 $count[j] \leftarrow count[j] + 1$;
15 **for** $i \leftarrow 1$ **to** $M$ **do**
16 **for** $j \leftarrow 1$ **to** $N$ **do**
17 $Mean[i][j] \leftarrow \frac{Sum[i][j]}{count[i]}$ ;
// End of the training phase

**Algorithm 2.  Pseudocode for MDEM in mean: Testing phase.**

**Data:** `Test[TestRows][N], Sum[M][N], count[M]`
**Result:** `Updated Mean[M][N] after testing`
1 **for** $i \leftarrow 1$ **to** $TestRows$ **do**
 // Testing phase
2 **for** $j \leftarrow 1$ **to** $N$ **do**
3 **for** $k \leftarrow 1$ **to** $M$ **do**
4 $tSum[k][j] \leftarrow Sum[k][j] + Test[i][j]$;
5 $tMean[k][j] \leftarrow \frac{tSum[k][j]}{count[k]+1}$ ;
6 **for** $k \leftarrow 1$ **to** $M$ **do**
7 $d[k] \leftarrow$ `Euclidean Distance(`$Mean[k]$`, `$tMean[k]$`)`;
8 $wd[k] \leftarrow d[k] \times count[k]$;
9 $Index \leftarrow$ `index at which` $wd[]$ `is minimum`;
10 $count[Index] \leftarrow count[Index] + 1$;
11 **for** $k \leftarrow 1$ **to** $N$ **do**
12 $Sum[Index][k] \leftarrow tSum[Index][k]$;
13 $Mean[Index][k] \leftarrow tMean[Index][k]$;

Next, we find the *Index* at which $wd[k], \forall k, 1 \le k \le M$ is minimized and this *Index* represents the class of this new testing instance under consideration. Then, we update the sum and mean of all attributes of *Index* class by the temporary

 

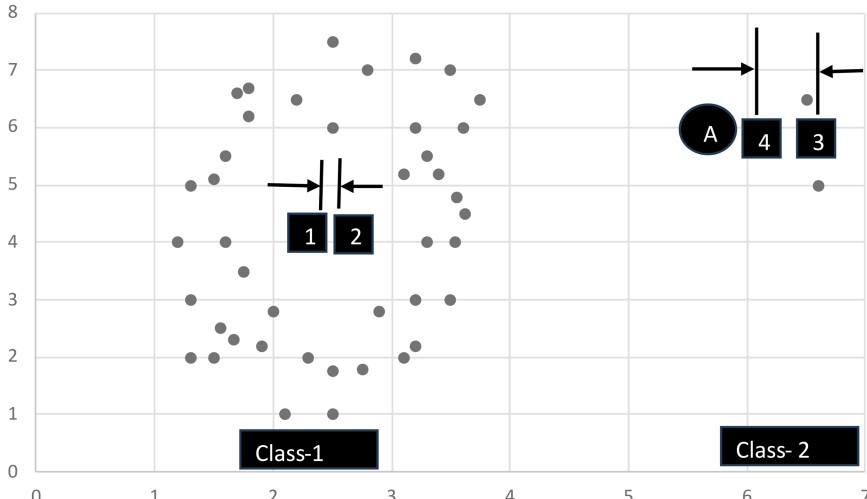

**Fig 1. Insertion of new data point.**

sum and temporary mean respectively. All other means and sums (other than that of *Index* class) are left unchanged before the commencement of the new iteration with a new testing instance.

### 3.2 MDEM in *n*-th central moment

The proposed algorithm involving higher order central moments intuitively uses the same idea as discussed in the previous subsection. In the training phase, we calculate the *n*-th central moment of every attribute of each class. Let us numerically describe the idea using 3rd order central moments ($n = 3$). Let us assume that the 5th attribute of the 2nd class has values of 5,10,15,8,20, which has a mean of 11.6. So, the 3rd central moment of the 5th attribute of the 2nd class is: $(1/5) \times ((5 - 11.6)^3 + (10 - 11.6)^3) + (15 - 11.6)^3 + (8 - 11.6)^3 + (20 - 11.6)^3)$ or, 58.75. So, after the training phase, we have the *n*-th central moment of each attribute of every class. At the testing phase, the new instance is temporarily included into all of the classes and the new temporary *n*-th central moment of each attribute of every class is calculated. Now, for each class $j, 1 \leq j \leq M$, both the temporary *n*-th central moment and existing *n*-th central moments are vectors of length *N*, where *N* is the number of attributes. Next, for each class *j*, we calculate the *N*-dimensional Euclidean distance between the temporary and existing *n*-th central moment. The distance thus calculated is then multiplied by the cardinality of the respective class in order to avoid the most densely populated class from engulfing every new test input.

To calculate *n*-th central moment in line with Eq 1, we have to have the *l*-th powered sum ($\forall l, 1 \leq l \leq n$) for each attribute of each class beforehand. These powered sums are generated in line [8–13] of Algorithm 3, where *Sum[l][j][k]* indicates *l*-th powered sum of the *k*-th attribute of the *j*-th class. Apart from the powered sums of each attribute of each class, we need to calculate the mean of each attribute of each class in order to determine *n*-th central moments for each attribute of each class in line with Eq 1. These means are generated in line [14–16] of Algorithm 3. Once we have the means and powered sums, we can calculate the *n*-th central moments according to formula given in Eq 1. To generate the expected values for any combination of *n*-th central moment and mean, we need to divide it by the cardinality of the respective class. The calculation of the *n*-th central moments are done in line [17–20] of Algorithm 3. This marks the end of the training phase of our algorithm.

After the end of the training phase, we have captured the values of *n*-th central moment of the *j–th* attribute ($\forall j, 1 \leq j \leq N$) of *i*-th class ($\forall i, 1 \leq i \leq M$) in *Moment[i][j]*. Next, we temporarily include every new training instance into every possible

**Algorithm 3.   Pseudocode for MDEM in *n*-th central moment: Training phase.**

**Data:** `TrainingData, TestingData`
**Result:** `Mean[M][N], Moment[M][N]`
1 $Mean[M][N] \leftarrow 0$ // `M is number of classes,` *N* `is number of attributes`
2 $tMean[M][N] \leftarrow 0$ // *tMean* `is temporary mean`
3 $Sum[n][M][N] \leftarrow 0$ // *n* `is number of moment considered`
4 $tSum[n][M][N] \leftarrow 0$;
5 $Train[TrainRows][N] \leftarrow$ `TrainingData`;
6 $Test[TestRows][N] \leftarrow$ `TestingData`;
7 $count[M] \leftarrow 0$;
 /* `Beginning of training phase` */
8 **for** $i \leftarrow 1$ **to** *TrainRows* **do**
9 **if** *i-th training instance belongs to class j* **then**
10 **for** $k \leftarrow 1$ **to** *N* **do**
11 **for** $l \leftarrow 1$ **to** *n* **do**
12 $Sum[l][j][k] \leftarrow Sum[l][j][k] + (Train[i][k])^l$;
13 $count[j] \leftarrow count[j] + 1$;

14 **for** $i \leftarrow 1$ **to** *M* **do**
15 **for** $j \leftarrow 1$ **to** *N* **do**
16 $Mean[i][j] \leftarrow \frac{Sum[1][i][j]}{count[i]}$ ;

17 **for** $i \leftarrow 1$ **to** *M* **do**
18 **for** $j \leftarrow 1$ **to** *N* **do**
19 **for** $k \leftarrow 0$ **to** *n* **do**
20 $Moment[i][j] \leftarrow Moment[i][j] + \binom{n}{k} \cdot \frac{Sum[n-k][i][j]}{count[i]} \cdot (-Mean[i][j])^k$;

 // `End of the training phase`

class and calculate the new temporary moment of each attribute of each class. This is done in line [1–9] of Algorithm 4. Next, we calculate the *N*-dimensional Euclidean distance between the temporary and existing *n*-th central moments and weight each such distance by the respective cardinality of each class. This is done in line [10–12] of Algorithm 4 and these weighted distances are preserved in *wd*[]. Next, we select the index value at which the weighted distance *wd*[] is minimized and this *index* value indicates the class to which the new test instance is classified by our algorithm. Once the class is fixed for the new instance, the existing moments, means and powered sums corresponding to that specific class are set to the temporary moment, temporary mean and temporary powered sums as calculated previously and the count for that specific class in increased by one. These are done in line [15–19] of Algorithm 4. For all other classes, existing means, moments and powered sums are left unchanged before the beginning of a new iteration for yet-to-be-classified test rows.

## 4 Time complexity analysis of the MDEM algorithm

In this section, we will analyze the time complexity of our proposed algorithm both in training and testing phase. We split our analysis into two parts: In first part, we determine the training and testing time complexity of MDEM in mean and in the second part, we analyze the training and testing time complexity of MDEM algorithms involving higher order central moments.

**Algorithm 4.    Pseudocode for MDEM in *n*-th central moment: Testing phase.**

**Data:** `Test[TestRows][N], Mean[M][N], Moment[M][N], Sum[n][M][N], count[M]`
**Result:** `Updated Mean, Moment, Sum, and count`

1  **for** $i \leftarrow 1$ **to** *TestRows* **do**
2 **for** $j \leftarrow 1$ **to** $N$ **do**
3 **for** $k \leftarrow 1$ **to** $M$ **do**
4 $tMean[k][j] \leftarrow \frac{Mean[k][j] \times count[k] + Test[i][j]}{count[k]+1}$ ;
5 **for** $l \leftarrow 1$ **to** $n$ **do**
6 $tSum[l][k][j] \leftarrow Sum[l][k][j] + (Test[i][j])^{l}$ ;
7 $tMoment[k][j] \leftarrow 0$ ;
8 **for** $l \leftarrow 0$ **to** $n$ **do**
9 $tMoment[k][j] \leftarrow$
 $tMoment[k][j] + \binom{n}{l} \cdot \frac{tSum[n-l][k][j]}{count[k]} \cdot (tMean[k][j])^{l}$ ;

10 **for** $j \leftarrow 1$ **to** $M$ **do**
11 $d[j] \leftarrow$ `Euclidean Distance(`$tMoment[j], Moment[j]$`)`;
12 $wd[j] \leftarrow d[j] \times count[j]$ ;
13 $Index \leftarrow$ `index at which` $wd[]$ `is minimum`;
14 $count[Index] \leftarrow count[Index] + 1$ ;
15 **for** $j \leftarrow 1$ **to** $N$ **do**
16 $Mean[Index][j] \leftarrow tMean[Index][j]$ ;
17 $Moment[Index][j] \leftarrow tMoment[Index][j]$ ;
18 **for** $l \leftarrow 1$ **to** $n$ **do**
19 $Sum[l][Index][j] \leftarrow tSum[l][Index][j]$ ;

## 4.1 Time complexity of MDEM in mean

In the training phase, we scan the training rows one by one, check the class of each instance, and if the class value is found to be *j*, then the value of $Sum[j][k], \forall k, 1 \leq k \leq N$ is increased by the amount of the *k*-th attribute value of the training instance under consideration. This is done in line [10–14] of Algorithm 1. So, time complexity of this step in the training phase is $O(N \times P)$, where $P$ is the number of training rows. As the number of attributes $N$ for a specific problem is fixed, the overall time complexity of this step is linear on the number of training rows. Once we have calculated the $Sum[][]$ for all the training rows, we can divide $Sum[i][j], \forall i, 1 \leq i \leq M, j, 1 \leq j \leq N$ by $count[i], \forall i, 1 \leq i \leq M$ to get the mean value of each attribute of each class. This is done in line [15–17] of Algorithm 1. Time needed in this step is $O(M \times N)$. For a specific problem, the number of classes ($M$) and the number of attributes ($N$) are fixed. Thus, the overall time complexity of the training phase is $O(P)$, where $P$ is the number of training instances.

In the testing phase, we temporarily include each training instance in every class and calculate the temporary mean of each attribute of each class, which can be done $O(M \times N)$ times, where $M$ is the number of classes, $N$ is the number of attributes. This is done in line [1–5] of Algorithm 2. Next, we calculate weighted and unweighted Euclidean distance between existing and temporary mean of each class, which is done in line [6–8] of Algorithm 2. Again, this can be done in $O(M)$ time. Next, we find the index value at which the weighted Euclidean distance as calculated above is minimized, which is calculated in $O(M)$ time (line 9 of Algorithm: 2). Finally, we update the sum and mean with the temporary sum and temporary mean of the attribute values of the respective class, which is done $O(N)$ time (line [11–13] of Algorithm 2). For a specific problem, $M$ and $N$ are fixed, which implies, every new instance can be classified in constant time.

## 4.2 Time complexity of MDEM in *n*-th central moment

In the training phase, we need to calculate $n$ ($n$ is the number of moments considered) number of powered sums as shown in line [8–13] of Algorithm 3. This can be done in $O(n)$ times. This step of calculating the powered sums needs to be repeated for each of the $N$ attributes of the training row under consideration. So, for each training row, we need to calculate $n \times N$ powered sums (line [10–12] of Algorithm 3). Steps as mentioned in line [10–12] are repeated for each training rows as well, and if there are $P$ number of training rows, then we need $O(n \times N \times P)$ number of operations in line [8–13] of Algorithm 3. For a specific problem, $n$ and $N$ are fixed beforehand. Thus, the time complexity of line [8–13] of training phase is linear on number of training rows, i.e., $O(P)$. Once the powered sums are generated, we can calculate the mean of each attribute of each class in $O(M \times N)$ time (line [14–16] of Algorithm 3) and the $n$-th central moment of each attribute of each class in $O(n \times M \times N)$ time (line [17–20] of Algorithm 3). As for a specific problem, the values of $n$, $M$ and $N$ are prefixed, the overall time complexity of the training phase is linear on the number of training instances, i.e., $O(P)$.

At the testing phase, we temporarily include every test instance into each of the possible $M$ classes and calculate the resulting temporary means, temporary powered sums and temporary moments (line [1–9] of Algorithm 4). For every test row, we need to calculate the temporary mean (line 4 of Algorithm 4), temporary powered sums (line [5–6] of Algorithm 4) and temporary moments (line [8–9] of Algorithm 4) for each attribute of each class, which can be calculated in $O(M \times N)$, $O(n \times M \times N)$ and $O(n \times M \times N)$ time respectively. As for a specific problem, the values of $n, M$ and $N$ are predetermined, the above steps can be completed in constant time for every new test instance. The next step in the testing phase involves calculating the Euclidean distance between the vectors of existing and temporary moments of every class (line [10–12] of Algorithm 4), which can be done in $O(M)$ time. Finding the index value at which such weighted distance $wd$ is minimum (line 18), can be done again in $O(M)$ time. Finally, updating the mean, temporary moment and temporary powered sums for the respective class can be done in $O(N)$, $O(N)$ and $O(n \times N)$ time (line [20–26]). As we have mentioned previously, the values of $n, M$ and $N$ are fixed beforehand for a specific problem, the overall time complexity of classifying a new testing instance is constant.

## 5 MDEM as a size-weighted minimum perturbation classifier

In this section, we will provide a theoretical foundation of the MDEM algorithms proposed in the current discourse. To do so, we invoke Minimum Perturbation Principle (MPP) that says, when we need to update a dynamic model to accommodate new instances, then the solution that minimally perturbs the system's initial structures and/or parameters should be selected. Minimum Perturbation Principle evaluates how small changes can influence classification and model output and is foundational in the areas of perceptual learning [14], adversarial machine learning [15,16], robustness testing [15], sensitivity-based feature selection [17] etc. In our context, we apply MPP with a view to minimizing weighted displacement in the existing n-th central moment of a particular class resulting from the inclusion of a new instance into that class. So, when we need to classify a new instance $x$, we first calculate the temporary moment $TM_i$ for each class due to inclusion of the new instance into class $i$ and measure how far away $TM_i$ is from the class's initial n-th central moment $M_i$, i.e., we estimate $|TM_i - M_i|$ for each class, weight such raw displacement with the class cardinality $|C_i|$ and choose the class for which such weighted displacement, $|TM_i - M_i| \times |C_i|$ is minimized. Except for the weighting factor, the idea is inspired from MPP. We now show that for both mean and n-th central moment of a class, such an unweighted difference between existing and new temporary moment is inversely proportional to class cardinality and as such, we need to multiply such unweighted difference with the cardinality of the respective class in order to get an unbiased estimate of the perturbation metric.

**Proposition 1:** When we add a new point to an existing set of points, then the displacement in the existing mean of the class resulting from such inclusion is inversely proportional to class size.

**Proof:** Let us assume we have $N$ number of points in a class and the points are given by $[x_1, x_2, x_3, ..., x_N]$. Let us also assume that the existing mean of the class is $\mu$, while the new mean after inclusion of a new point $x_{N+1}$ is $\mu'$. Thus,

$$\mu' = \frac{N \times \mu + x_{N+1}}{N+1}$$

Therefore, the change in mean $\Delta\mu$ as a result of inclusion of point $x_{N+1}$ is given by the following:

$$\Delta\mu = \mu' - \mu = \frac{N \times \mu + x_{N+1}}{N+1} - \mu = \frac{x_{N+1} - \mu}{N+1}$$

From the above expression, we can see that the displacement $\Delta\mu$ is inversely proportional to the new class cardinality (N+1).

**Proposition 2:** When we add a new point to an existing set of points, then the displacement in the existing n-th central moment of the class resulting from such inclusion is inversely proportional to class size.

**Proof:** Suppose we have a dataset with $N$ points: $[x_1, x_2, ..., x_N]$. The mean of these points is given by the following:

$$\bar{x}_N = \frac{1}{N} \sum_{i=1}^{N} x_i$$

By definition, the *n*-th central moment of the class is given by:

$$\mu_n^{(N)} = \frac{1}{N} \sum_{i=1}^{N} (x_i - \bar{x}_N)^n$$

Now, let us separately analyze the scenario how the mean and n-th central moment of the class change due to inclusion of a new point $x_{N+1}$ into it.

• **New mean after inclusion of a new point**

   Due to inclusion of $x_{N+1}$, the new mean becomes:

$$\bar{x}_{N+1} = \frac{N\bar{x}_N + x_{N+1}}{N+1}$$

$$\bar{x}_{N+1} = \bar{x}_N + \frac{\bar{x}_{N+1} - \bar{x}_N}{N+1}$$

   Now let us assume that the distance between the new point and the existing mean is $\delta$, i.e., $\delta = x_{N+1} - \bar{x}_N$. If we substitute the value of $\delta$ into the above expression of new mean, we get:

$$\bar{x}_{N+1} = \bar{x}_N + \frac{\delta}{N+1} \tag{5}$$

- **New central moment after inclusion of a new point**

  The new $n$-th central moment is:

  $$\mu_n^{(N+1)} = \frac{1}{N+1}\left(\sum_{i=1}^{N}(x_i - \bar{x}_{N+1})^n + (x_{N+1} - \bar{x}_{N+1})^n\right)$$

  Now, we substitute the value of $\bar{x}_{N+1}$ from Eq 5 inside the summation sign of the above expression. Doing so, the above expression turns out to be:

  $$\mu_n^{(N+1)} = \frac{1}{N+1}\left(\sum_{i=1}^{N}(x_i - \bar{x}_N - \frac{\delta}{N+1})^n + (x_{N+1} - \bar{x}_{N+1})^n\right)$$

  Let us asume, $h = \frac{\delta}{N+1}$. This will reduce the above expression into the following.

  $$\mu_n^{(N+1)} = \frac{1}{N+1}\left(\sum_{i=1}^{N}((x_i - \bar{x}_N) + (-h))^n + (x_{N+1} - \bar{x}_{N+1})^n\right) \qquad (6)$$

  Then by expanding the term inside the summation sign by binomial expansion, we get:

  $$((x_i - \bar{x}_N) + (-h))^n = \sum_{k=0}^{n}\binom{n}{k}(x_i - \bar{x}_N)^{n-k}(-h)^k$$

  Substituting the value of $(x_i - \bar{x}_N) + (-h))^n$ from the above equation into Eq 6 yields:

  $$\mu_n^{(N+1)} = \frac{1}{N+1}\left(\sum_{i=1}^{N}\sum_{k=0}^{n}\binom{n}{k}(x_i - \bar{x}_N)^{n-k}(-h)^k + (x_{N+1} - \bar{x}_{N+1})^n\right)$$

  As the indices of both sums range over finite sets, we can use the property of associativity and commutativity to interchange the ordering of the summations as follows:

  $$\mu_n^{(N+1)} = \frac{1}{N+1}\left(\sum_{k=0}^{n}\sum_{i=1}^{N}\binom{n}{k}(x_i - \bar{x}_N)^{n-k}(-h)^k + (x_{N+1} - \bar{x}_{N+1})^n\right)$$

  At this stage, we may recall that:

  $$\sum_{i=1}^{N}(x_i - \bar{x}_N)^{n-k} = N\mu_{n-k}^{(N)}$$

  Substituting this into the above expression, we get:

  $$\mu_n^{(N+1)} = \frac{1}{N+1}\left(\sum_{k=0}^{n}\binom{n}{k}N\mu_{n-k}^{(N)}(-h)^k + (x_{N+1} - \bar{x}_{N+1})^n\right) \qquad (7)$$

Now, we rewrite the term outside the summation, i.e., $(x_{N+1} - \bar{x}_{N+1})^n$ in a manner that serves our purpose. So, we substitute the value $\bar{x}_{N+1}$ from Eq 5 into the above expression, which results into:

$$(x_{N+1} - \bar{x}_{N+1})^n = (x_{N+1} - \bar{x}_N - \frac{\delta}{N+1})^n = (\delta - \frac{\delta}{N+1})^n = \delta^n \left(\frac{N}{N+1}\right)^n$$

Now, subsituting the value of $(x_{N+1} - \bar{x}_{N+1})^n$ from the above expression into Eq 7 yields:

$$\mu_n^{(N+1)} = \frac{N}{N+1} \left[ \sum_{k=0}^{n} \binom{n}{k}(-h)^k \mu_{n-k}^{(N)} + \delta^n \left(\frac{N}{N+1}\right)^n \right] \tag{8}$$

- **Displacement in n-th central moment**
  The displacement in the n-th central moment due to inclusion of a new point is given by:

$$\Delta\mu_n = \mu_n^{(N+1)} - \mu_n^{(N)}$$

Now, substituting the value of $\mu_n^{N+1}$ from Eq 8 into the above expression, we get:

$$\Delta\mu_n = \frac{N}{N+1} \left[ \sum_{k=0}^{n} \binom{n}{k}(-h)^k \mu_{n-k}^{(N)} + \delta^n \left(\frac{N}{N+1}\right)^n \right] - \mu_n^{(N)} \tag{9}$$

Now, the scaling factor $\frac{N}{N+1}$ fades out to 1 for any practical application as the value of N grows. Moreover, as we may recall here that the value of $\delta$ is defined as the difference between the new point and the existing mean, i.e., $x_{N+1} - \bar{x}_N$. Thus, $\delta$ is independent of our decision making process and is given beforehand. Additionally, when we are estimating n-th central moment involving $(N + 1)$ instances, all the lower order moments involving $N$ number of elements are already given, i.e., all the $\mu_{n-k}^{(N)}, \forall k, 0 \le k \le n$ are estimated and do not change. So, the only variable thing that remains in the expression relating to moment difference is a power series of $h$ and as we have discussed earlier that $h$ is inversely proportional to class cardinality as it is defined as $\frac{\delta}{N+1}$. As $h$ is inversely proportional to class cardinality, higher order terms of $h$ in the displacement equation will increasingly have lower contribution to the measured displacement as $N$ grows. Thus, we can assume that the stated displacement measure is inversely proportional to the first power of $h$, i.e., $O(\frac{\delta}{N+1})$, or more consicely, $O(\frac{1}{N+1})$, as $\delta$ is given beforehand and is independent of our selection criteria. This completes the proof.

## 6 A sufficient condition for optimality under minimum perturbation criteria

Suppose we need to classify a new instance $x$ and let the true class of $x$ be $c^*$. To do so, we need to calculate the displacement of the existing moment of each class caused by the temporary inclusion of the new instance $x$ into that particular class. For class $c$, let this empirical displacement be denoted by $\widehat{D}_c(x)$. We will include the new instance $x$ into the class for which the estimated $\widehat{D}_c(x)$ is minimum, i.e.,

$$\hat{c} = \arg\min_c \widehat{D}_c(x)$$

One important thing to note here is that we are estimating empirical displacement, i.e., $\widehat{D}_c(x)$, which may be different from true population displacement $D_c(x)$. This implies that the true population classifier should choose the label of the new

instance as follows:

$$c^* = \arg\min_c D_c(x)$$

As the true class is $c^*$, $D_{c^*}(x)$ will be minimum and all other $D_c(x)$ will be greater than $D_{c^*}(x)$. Let us now define a margin $m(x)$ that represents the displacement between $D_c^*(x)$ and the second best option, i.e.,

$$m(x) = \min_{c \neq c^*}\Big(D_c(x) - D_{c^*}(x)\Big).$$

We now use this definition of margin to establish a sufficient condition for optimality under minimum perturbation criteria of our proposed approach.

**Proposition 3:** If the difference between empirical and true population displacement for all the classes is bounded by $m(x)/2$, i.e., $\big|\widehat{D}_c(x) - D_c(x)\big| \leq \frac{m(x)}{2}$, then MDEM will always choose the correct class under minimum perturbation criteria.

**Proof:**

From the assumption, we have $\big|\widehat{D}_c(x) - D_c(x)\big| \leq \frac{m(x)}{2}$ for every class $c$. This implies:

$$-\frac{m(x)}{2} \leq \widehat{D}_c(x) - D_c(x) \leq \frac{m(x)}{2}. \tag{10}$$

For the optimal class $c^*$ under minimum perturbation criteria, we use the inequality involving the second and third term of Eq 10, which entails:

$$\widehat{D}_{c^*}(x) \leq D_{c^*}(x) + \frac{m(x)}{2}. \tag{11}$$

On the other hand, for any class $c$, taking first two terms of Eq 10, we have:

$$\begin{aligned}
\widehat{D}_c(x) &\geq D_c(x) - \frac{m(x)}{2} \\
&\geq D_{c^*}(x) + m(x) - \frac{m(x)}{2} \qquad \text{(substituting from margin definition)} \\
&= D_{c^*}(x) + \frac{m(x)}{2} \\
&\geq \widehat{D}_{c^*}(x) \qquad \text{(using Eq 11)}
\end{aligned}$$

Thus, $\widehat{D}_c(x) \geq \widehat{D}_{c^*}(x)$ for all $c$.

This means, for any arbitrary class $c$, the estimated empirical distance is always greater than the estimated empirical distance of the true class, provided the condition mentioned in proposition 3 is satisfied. As the estimated empirical distance is minimum for the true class $c^*$, MDEM always choose $c^*$ as the labelling class for a new instance $x$ under minimum perturbation criteria. This completes the proof. □

## 7 Methodology

To begin with, we apply various data preprocessing techniques, e.g., identification and replacement of missing values, detection and removal of outliers using Inter Quartile Range (IQR) filter and feature scaling techniques to normalize the data within the range [0-1]. Processed data are then fed into various machine learning algorithms including MDEM in mean, MDEM in variance (2nd central moment), MDEM in 3rd central moment, MDEM in 4th central moment, KNN with 3, 5, 7 neighbors and NN with 1, 2, 3 hidden layers each comprising 5 neurons with 50, 100 and 150 epochs. We use Eqs 2,

3 and 4 to calculate 2nd, 3rd and 4th central moments used in our proposed analysis based on Minimum Displacement in Existing Moment (MDEM) technique.

Here, we only consider first raw moment as well as 2nd, 3rd and 4th order central moments as they capture some specific attributes of the underlying distribution, namely, mean, variance, skewness and kurtosis respectively. It is to be noted in this regard that moments, higher in order than the 4th order moments, are known as hyper-skewness (odd ones) and hyper-kurtosis (even one) that are rarely used in statistical analysis and only capture fine details of skewness and kurtosis respectively.

For the purpose of this analysis, we randomly split our dataset into $k$ equally sized folds. We then use some $(k-1)$ folds to train our model and the rest 1 fold is used to analyze the performance of the trained model, which is popularly known as k-fold cross validation technique. The choice of $k$ is rather arbitrary and there exists bias-variance trade-offs associated with the choice of $k$ in k-fold cross validation [18]. One of the preferred choices for $k$ is 5, as it is shown to yield test error estimates that do not suffer either from excessively high bias or from extreme variances [18]. So, we check the performance of different machine learning algorithms using 5-fold cross validation technique. Apart from that, we also compare the performance of different algorithms under 2, 3 and 7-fold cross validation also.

While k-Fold cross validation technique may perform well for balanced data, its performance deteriorates once it is used handle imbalanced or skewed data [19]. As we will mention later in the data section, the data used in our present analysis are skewed to some degree, e.g., we have unequal number of observations in each class. To overcome the hurdles faced by k-fold cross validation techniques to classify imbalanced data, we use stratified k-fold cross validation, which intends to solve the problem of imbalanced data to some extent. In stratified k-fold cross validation, the folds are generated by preserving the relative percentage of each class. Like k-fold, we use 2, 3, 5 and 7 number of folds in our stratified k-fold cross validation technique to analyze the performance of different algorithms.

## 8 Description of data

We collect Pima Indian Diabetes (PID) dataset originally developed by the National Institute of Diabetes and Digestive and Kidney Diseases (NIDDK), which is part of the United States' National Institutes of Health. The Pima Indians are native American people, who traditionally lived along the Gila and Salt rivers in Arizona, United States and can now be found in various parts of Arizona, US and Mexico [20]. This group has a high prevalence of diabetes among its members and diabetes research around the Pima Indians are often considered to be significant and representative of the global health [21]. The PID dataset comprises records of some 768 females having age 21 and above from the Pima Indian population and is widely considered as a benchmark dataset in diabetes research [22]. 08 (eight) attributes, namely, number of pregnancies, plasma glucose concentration in an oral glucose tolerance test, diastolic blood pressure, triceps skin fold thickness, serum insulin level, BMI, diabetes pedigree function, age and a class variable representing the prevalence of diabetes in the respective individual are recorded. Descriptive statistics of the PID dataset are presented in Table 1.

As can be seen from Table 1, out of 768 records in PID dataset, 500 are non-diabetic, while the rest 268 are diabetic.

Apart from PID dataset, we also use Wisconsin breast cancer dataset that comprise 569 records each containing 30 features [23]. The mentioned feature values are extracted from digitized images of different fine needle aspirates (FNA) of breast masses. To be precise, there are 10 real valued features, namely, radius (mean of distances from center to points on the perimeter), texture (standard deviation of gray-scale values), perimeter, area, smoothness (local variation in radius lengths), compactness (squared perimeter/area - 1.0), concavity (severity of concave portions of the contour), concave points (number of concave portions of the contour), symmetry and fractal dimension (coastline approximation - 1). From these 10 real valued features, a total of 30 features are synthesized that represent mean, standard error and worst (mean of the three largest values) of the feature values. There is one output value, which represents whether the extracted breast mass is benign or malignant. In the main dataset, each record also contains the patient ID number, which is not contextual in the current analysis. Descriptive statistics of the records are presented in Table 2.

**Table 1**. Descriptive statistics of Pima Indian Diabetes (PID) dataset.

| No | Attribute description | Attribute type | Average value |
|---|---|---|---|
| 1 | Number of times pregnant | Numeric | 3.85 |
| 2 | Plasma glucose concentration in 2h in an oral glucose tolerance test | Numeric | 120.89 |
| 3 | Diastolic blood pressure (mm HG) | Numeric | 69.11 |
| 4 | Triceps skin fold thickness (mm) | Numeric | 20.54 |
| 5 | Serum insulin level ($\mu IU/mL$) | Numeric | 79.80 |
| 6 | Body Mass Index (BMI) ($kg/m^2$) | Numeric | 31.99 |
| 7 | Diabetes Pedigree Function | Numeric | 0.47 |
| 8 | Age (years) | Numeric | 33.24 |
| 9 | Output variable | Nominal; 1: Diabetic; 0: Non-Diabetic | 268 Diabetic; 500 non-diabetic |

**Table 2**. Descriptive statistics of Wisconsin breast cancer dataset.

| No | Attribute name | Attribute type | Average values |
|---|---|---|---|
| 1 | Radius mean | Numeric | 14.12729174 |
| 2 | Texture mean | Numeric | 19.28964851 |
| 3 | Perimeter mean | Numeric | 91.96903339 |
| 4 | Area mean | Numeric | 654.8891037 |
| 5 | Smoothness mean | Numeric | 0.096360281 |
| 6 | Compactness mean | Numeric | 0.104340984 |
| 7 | Concavity mean | Numeric | 0.088799316 |
| 8 | Concave point mean | Numeric | 0.048919146 |
| 9 | Symmetry mean | Numeric | 0.181161863 |
| 10 | Fractal dimension mean | Numeric | 0.06279761 |
| 11 | Radius SE | Numeric | 0.405172056 |
| 12 | Texture SE | Numeric | 1.216853427 |
| 13 | Perimeter SE | Numeric | 2.866059227 |
| 14 | Area SE | Numeric | 40.33707909 |
| 15 | Smoothness SE | Numeric | 0.007040979 |
| 16 | Compactness SE | Numeric | 0.025478139 |
| 17 | Concavity SE | Numeric | 0.031893716 |
| 18 | Concave point SE | Numeric | 0.011796137 |
| 19 | Symmetry SE | Numeric | 0.020542299 |
| 20 | Fractal dimension SE | Numeric | 0.003794904 |
| 21 | Radius worst | Numeric | 16.26918981 |
| 22 | Texture worst | Numeric | 25.6772232 |
| 23 | Perimeter worst | Numeric | 107.2612127 |
| 24 | Area worst | Numeric | 880.5831283 |
| 25 | Smoothness worst | Numeric | 0.132368594 |
| 26 | Compactness worst | Numeric | 0.254265044 |
| 27 | Concavity worst | Numeric | 0.272188483 |
| 28 | Concave point worst | Numeric | 0.114606223 |
| 29 | Symmetry worst | Numeric | 0.290075571 |
| 30 | Fractal dimension worst | Numeric | 0.083945817 |
| 31 | Output | Categorical | B: 357 Benign, M: 212 Malignant |

As can be seen from Table 2, out of 569 records in Wisconsin breast cancer dataset, 357 breast masses are benign, while the rest 212 are malignant.

## 9 Data preprocessing

Data preprocessing is essential before feeding any data into machine learning algorithms as it helps build a better machine learning model with greater accuracy. Data preprocessing in our current analysis involves identification and replacement of missing values, detection and removal of the outliers and normalization.

## 9.1 Identification and replacement of missing values

In this step, we identify the missing values in the PID dataset and replace them with the corresponding mean values. It has been observed that three attributes, namely, number of pregnancies, diabetes pedigree function and age have no missing values, i.e., we have all 768 values for these three attributes. For the other attributes, there are some missing values. To be precise, plasma glucose level, blood pressure, triceps skin thickness, insulin level and BMI have 5, 35, 227, 374 and 11 missing values. We replace the missing values with their respective mean values for the sake of our current analysis.

On the other hand, there is no missing value in Wisconsin breast cancer dataset.

## 9.2 Detection and removal of outliers

In this step, we identify and remove outliers from the PID dataset. Outliers are simply data points that differ significantly from all other data points under consideration and they may arise due to variability in the measurement, experimental errors etc. So, in this step, we remove the outliers from the dataset lest we run the risk of building an over-fitted model that may perform well for training data but, behaves poorly for new test data. We use Inter Quartile Range (IQR) filter to detect and remove outliers from our dataset. After applying the IQR filter on our data, we have observed that there are 49 outliers and 719 normal records. We remove these 49 outliers from our analysis and continue with the remaining 719 observations.

On the other hand, applying IQR filter on Wisconsin breast cancer data, we find that there are 55 outliers and 10 extreme values. We remove them at this step as part of the data cleansing process.

## 9.3 Feature scaling

Feature scaling is a data preprocessing technique intended to standardize the attribute values within a permissible range. In our present analysis, we have heterogeneous attribute values that differ significantly from one another in their orders of magnitude. For example, the number of pregnancies in the PID dataset varies between [0-17], while data on plasma glucose level swings between an even larger range of [44-199]. So, if we do not normalize the data, then the machine learning algorithms will tend to provide higher weightages to plasma glucose level than number of pregnancies, which is not intended. Thus, by normalizing all the attribute values within the range [0-1], we can build a better machine learning model. We use unsupervised normalization filter in Weka to normalize the attribute values within the range of [0-1].

Same is applicable for Wisconsin breast cancer data as well. Like the PID dataset, the features in Wisconsin breast cancer data also vary widely in their magnitudes. For example, the average value of perimeter mean is 91.97, while the average value of concavity mean is only 0.09. So, it is better to normalize all the features within [0-1] range before feeding them into any learning algorithm. Like PID dataset, we use unsupervised normalization filter in Weka to cast the attribute values within [0-1] range.

## 10 Results

### 10.1 PID dataset

We run 19 different machine learning algorithms, namely, MDEM (in 4 variants), Logistic Regression (LR), Random Forest (RF), Support Vector Machine (SVM), K-Nearest Neighbor (KNN) (in 3 variants), Neural Network (NN) (in 9 variants) and note down their performances based upon their ability to properly classify PID dataset. Apart from the accuracy scores, we also calculate the confusion matrix to compare the overall performances of of different algorithms. We use both k-fold and stratified k-fold cross validation techniques to analyze the performances of different algorithms. Apart from calculating the accuracy scores of different models, we also estimate their respective confusion matrices. Results under k-fold cross validation techniques are presented in Tables 3, 4 and 5, while the results under stratified k-fold cross validation techniques are presented in Tables 6, 7 and 8.

**Table 3**. Performance analyses of different algorithms using k-fold cross validation technique for PID dataset.

| Algorithm | 2-Fold | 3-Fold | 5-Fold | 7-Fold |
|---|---|---|---|---|
| MDEM (Mean) | 73.43% | 73.57% | 73.44% | 73.16% |
| MDEM (Variance) | 70.51% | 70.37% | 70.37% | 70.93% |
| MDEM (3rd Central Moment) | 68.00% | 68.00% | 67.74% | 67.46% |
| MDEM (4th Central Moment) | 68.29% | 67.17% | 65.23% | 64.81% |
| Logistic Regression (LR) | 74.83% | 77.47% | 76.92% | 77.34% |
| Random Foresh (RF) | 75.94% | 76.22% | 76.92% | 76.49% |
| Support Vector Machine (SVM) | 76.63% | 77.19% | 77.06% | 77.48% |
| K Nearest Neighbor (KNN) (n = 3) | 69.82% | 72.05% | 71.49% | 71.78% |
| K Nearest Neighbor (KNN) (n = 5) | 70.38% | 71.91% | 73.30% | 74.28% |
| K Nearest Neighbor (KNN) (n = 7) | 72.32% | 72.74% | 72.05% | 73.02% |
| Neural Network (Hidden Layer = 1, Epoch = 50) | 67.04% | 66.34% | 67.05% | 68.57% |
| Neural Network (Hidden Layer = 1, Epoch = 100) | 66.48% | 70.93% | 69.68% | 73.16% |
| Neural Network (Hidden Layer = 1, Epoch = 150) | 70.65% | 72.19% | 72.32% | 75.82% |
| Neural Network (Hidden Layer = 2, Epoch = 50) | 67.18% | 67.88% | 68.02% | 70.52% |
| Neural Network (Hidden Layer = 2, Epoch = 100) | 71.62% | 75.39% | 74.41% | 74.55% |
| Neural Network (Hidden Layer = 2, Epoch = 150) | 73.15% | 76.22% | 75.81% | 76.64% |
| Neural Network (Hidden Layer = 3, Epoch = 50) | 67.73% | 72.33% | 71.22% | 73.58% |
| Neural Network (Hidden Layer = 3, Epoch = 100) | 73.01% | 75.11% | 76.23% | 76.22% |
| Neural Network (Hidden Layer = 3, Epoch = 150) | 74.55% | 72.75% | 76.08% | 76.92% |

From Table 3, we can see that MDEM in mean, variance, 3rd and 4th central moment have obtained accuracy scores of 73.43%, 70.51%, 68.00% and 68.29% respectively under 2-fold cross validation technique. The accuracies obtained by Logistic Regression (LR), Random Forest (RF) and Support Vector Machine (SVM) are found to be 74.83%, 75.94% and 76.63%. On the other hand, KNN algorithms with 3, 5, 7 neighbors have accuracies of 69.82%, 70.38% and 72.32% respectively. Moreover, the performance of the NN varies somewhere between 66.48% to 74.55% depending upon the number of hidden layers and epochs. Best for the NN under current consideration, i.e., accuracy score of 74.55% is obtained for NN with 3 hidden layers with 150 epochs, while the worst is obtained for NN with 1 hidden layer in 100 epochs. As can be seen from Table 3, MDEM in mean runs better than KNN with 3, 5, 7 neighbors and 8 (eight) of the NN models under consideration in 2-fold cross validation technique. The best algorithm under 2-fold cross validation is found to be Support Vector Machine (SVM) having a run time accuracy of 76.63%. So, the performances of MDEM in 04 different variants are within the range of [89.12% − 95.82%] of the best algorithm under consideration. The results are graphically presented in Fig 2.

Moreover, as can be seen from Table 4, the precision scores of different MDEM algorithms vary between 0.54–0.58. For 2-fold cross validation, the best precision score of 0.73 is obtained for NN with 1 hidden layer in 100 epochs. So, precision scores of MDEM algorithms are within [73.97% − 79.45%] of the best algorithm. On the other hand, recall scores of MDEM algorithms are found within the range of [0.37–0.68]. It is to be noted in this regard that MDEM in mean has obtained the best recall score of 0.68 amongst all the algorithm considered. With a high recall score, MDEM algorithms have obtained F1 score and MCC of [0.44–0.63] and [0.24–0.43] respectively. From Table 5, we can see that MDEM in mean has obtained the highest F1-score of 0.63, while its MCC of 0.43 is only next to that of SVM and RF.

Under 3-fold cross validation, the accuracies of MDEM in mean, variance, 3rd and 4th central moments are 73.57%, 70.37%, 68.00% and 67.17% respectively. The best algorithm under 3-fold cross validation is Logistic Regression (LR) having an accuracy of 77.47%. So, the performances of different variants of MDEM are within the range of [86.78% − 94.97%] of the best algorithm for 3-fold. Details are given in Fig 2. Moreover, from Table 5, we can see that the F1 score and MCC of different MDEM algorithms vary within the range of [0.46–0.64] and [0.24–0.43] respectively. F1 score of MDEM in mean is the highest amongst the all 19 algorithms considered, while its MCC is only next to that of SVM and RF.

**Table 4.** Precision and recall scores of different algorithms using k-fold cross validation technique for PID dataset.

| Algorithm | 2-Fold | 3-Fold | 5-Fold | 7-Fold |
|---|---|---|---|---|
| Precision: | | | | |
| MDEM (Mean) | 0.58 | 0.59 | 0.59 | 0.58 |
| MDEM (Variance) | 0.55 | 0.55 | 0.55 | 0.56 |
| MDEM (3rd Central Moment) | 0.54 | 0.53 | 0.52 | 0.51 |
| MDEM (4th Central Moment) | 0.54 | 0.52 | 0.49 | 0.48 |
| Logistic Regression (LR) | 0.69 | 0.73 | 0.72 | 0.74 |
| Random Foresh (RF) | 0.68 | 0.66 | 0.68 | 0.68 |
| Support Vector Machine (SVM) | 0.72 | 0.74 | 0.73 | 0.74 |
| K Nearest Neighbor (KNN) (n = 3) | 0.55 | 0.59 | 0.59 | 0.60 |
| K Nearest Neighbor (KNN) (n = 5) | 0.57 | 0.59 | 0.63 | 0.63 |
| K Nearest Neighbor (KNN) (n = 7) | 0.61 | 0.61 | 0.60 | 0.61 |
| Neural Network (Hidden Layer = 1, Epoch = 50) | 0.19 | 0.65 | 0.55 | 0.63 |
| Neural Network (Hidden Layer = 1, Epoch = 100) | 0.73 | 0.46 | 0.58 | 0.49 |
| Neural Network (Hidden Layer = 1, Epoch = 150) | 0.65 | 0.63 | 0.65 | 0.60 |
| Neural Network (Hidden Layer = 2, Epoch = 50) | 0.29 | 0.21 | 0.58 | 0.55 |
| Neural Network (Hidden Layer = 2, Epoch = 100) | 0.67 | 0.66 | 0.53 | 0.71 |
| Neural Network (Hidden Layer = 2, Epoch = 150) | 0.64 | 0.63 | 0.64 | 0.70 |
| Neural Network (Hidden Layer = 3, Epoch = 50) | 0.63 | 0.59 | 0.39 | 0.62 |
| Neural Network (Hidden Layer = 3, Epoch = 100) | 0.57 | 0.65 | 0.64 | 0.74 |
| Neural Network (Hidden Layer = 3, Epoch = 150) | 0.65 | 0.62 | 0.53 | 0.70 |
| Recall: | | | | |
| MDEM (Mean) | 0.68 | 0.70 | 0.70 | 0.70 |
| MDEM (Variance) | 0.62 | 0.64 | 0.65 | 0.66 |
| MDEM (3rd Central Moment) | 0.35 | 0.38 | 0.40 | 0.41 |
| MDEM (4th Central Moment) | 0.37 | 0.43 | 0.45 | 0.47 |
| Logistic Regression (LR) | 0.44 | 0.51 | 0.51 | 0.51 |
| Random Foresh (RF) | 0.56 | 0.57 | 0.57 | 0.58 |
| Support Vector Machine (SVM) | 0.49 | 0.50 | 0.51 | 0.52 |
| K Nearest Neighbor (KNN) (n = 3) | 0.48 | 0.53 | 0.49 | 0.50 |
| K Nearest Neighbor (KNN) (n = 5) | 0.47 | 0.51 | 0.51 | 0.56 |
| K Nearest Neighbor (KNN) (n = 7) | 0.47 | 0.52 | 0.50 | 0.53 |
| Neural Network (Hidden Layer = 1, Epoch = 50) | 0.04 | 0.18 | 0.14 | 0.07 |
| Neural Network (Hidden Layer = 1, Epoch = 100) | 0.18 | 0.24 | 0.36 | 0.28 |
| Neural Network (Hidden Layer = 1, Epoch = 150) | 0.32 | 0.42 | 0.44 | 0.41 |
| Neural Network (Hidden Layer = 2, Epoch = 50) | 0.15 | 0.10 | 0.31 | 0.36 |
| Neural Network (Hidden Layer = 2, Epoch = 100) | 0.26 | 0.51 | 0.39 | 0.49 |
| Neural Network (Hidden Layer = 2, Epoch = 150) | 0.47 | 0.55 | 0.55 | 0.54 |
| Neural Network (Hidden Layer = 3, Epoch = 50) | 0.18 | 0.45 | 0.28 | 0.49 |
| Neural Network (Hidden Layer = 3, Epoch = 100) | 0.43 | 0.58 | 0.54 | 0.58 |
| Neural Network (Hidden Layer = 3, Epoch = 150) | 0.49 | 0.55 | 0.44 | 0.58 |

For 5-fold cross validation, the accuracies of different MDEM algorithms are 73.44%, 70.37%, 67.74% and 65.23% respectively. As can be seen from Table 3, the best algorithm under 5-fold cross validation technique is identified to be Support Vector Machine (SVM) with a run time accuracy of 77.06%. So, MDEM algorithms run within the range of [87.54% − 95.30%] of the best algorithm under 5-fold cross validation. Details are given in Fig 2. Additionally, from Table 5, we can find that the F1 score and MCC of different MDEM algorithms lie within [0.46–0.64] and [0.21–0.44] respectively. From Table 5, it is also evident that F1 score of MDEM in mean is the best amongst the all 19 algorithms considered, while its MCC is only next to that of LR, RF and SVM.

For 7-fold cross validation, the accuracies of MDEM in mean, variance, 3rd and 4th central moments are 73.16%, 70.93%, 67.46% and 64.81% respectively. The best algorithm for 7-fold cross validation is found to be Support Vector Machine (SVM) with a run time accuracy of 77.48%. So, different MDEM algorithms run within the range of [83.65% − 94.42%] of the best algorithm under 7-fold cross validation technique. Detailed results are graphically presented in Fig 2.

**Table 5**. **F1 scores and MCC of different algorithms using k-fold cross validation technique for PID dataset.**

| Algorithm | 2-Fold | 3-Fold | 5-Fold | 7-Fold |
|---|---|---|---|---|
| F1 Score: | | | | |
| MDEM (Mean) | 0.63 | 0.64 | 0.64 | 0.63 |
| MDEM (Variance) | 0.58 | 0.59 | 0.59 | 0.60 |
| MDEM (3rd Central Moment) | 0.43 | 0.44 | 0.45 | 0.45 |
| MDEM (4th Central Moment) | 0.44 | 0.46 | 0.46 | 0.46 |
| Logistic Regression (LR) | 0.54 | 0.60 | 0.59 | 0.60 |
| Random Foresh (RF) | 0.61 | 0.61 | 0.62 | 0.62 |
| Support Vector Machine (SVM) | 0.58 | 0.59 | 0.59 | 0.60 |
| K Nearest Neighbor (KNN) (n = 3) | 0.51 | 0.56 | 0.53 | 0.54 |
| K Nearest Neighbor (KNN) (n = 5) | 0.51 | 0.55 | 0.56 | 0.59 |
| K Nearest Neighbor (KNN) (n = 7) | 0.53 | 0.56 | 0.54 | 0.56 |
| Neural Network (Hidden Layer = 1, Epoch = 50) | 0.06 | 0.28 | 0.22 | 0.12 |
| Neural Network (Hidden Layer = 1, Epoch = 100) | 0.29 | 0.31 | 0.43 | 0.35 |
| Neural Network (Hidden Layer = 1, Epoch = 150) | 0.43 | 0.50 | 0.52 | 0.48 |
| Neural Network (Hidden Layer = 2, Epoch = 50) | 0.20 | 0.13 | 0.39 | 0.43 |
| Neural Network (Hidden Layer = 2, Epoch = 100) | 0.35 | 0.57 | 0.44 | 0.56 |
| Neural Network (Hidden Layer = 2, Epoch = 150) | 0.54 | 0.58 | 0.58 | 0.61 |
| Neural Network (Hidden Layer = 3, Epoch = 50) | 0.27 | 0.50 | 0.32 | 0.54 |
| Neural Network (Hidden Layer = 3, Epoch = 100) | 0.49 | 0.61 | 0.58 | 0.61 |
| Neural Network (Hidden Layer = 3, Epoch = 150) | 0.56 | 0.58 | 0.48 | 0.63 |
| MCC: | | | | |
| MDEM (Mean) | 0.43 | 0.43 | 0.44 | 0.43 |
| MDEM (Variance) | 0.36 | 0.36 | 0.37 | 0.38 |
| MDEM (3rd Central Moment) | 0.23 | 0.24 | 0.23 | 0.23 |
| MDEM (4th Central Moment) | 0.24 | 0.24 | 0.21 | 0.21 |
| Logistic Regression (LR) | 0.40 | 0.46 | 0.46 | 0.47 |
| Random Foresh (RF) | 0.44 | 0.44 | 0.46 | 0.46 |
| Support Vector Machine (SVM) | 0.45 | 0.46 | 0.46 | 0.47 |
| K Nearest Neighbor (KNN) (n = 3) | 0.30 | 0.36 | 0.34 | 0.35 |
| K Nearest Neighbor (KNN) (n = 5) | 0.31 | 0.35 | 0.38 | 0.41 |
| K Nearest Neighbor (KNN) (n = 7) | 0.35 | 0.37 | 0.35 | 0.37 |
| Neural Network (Hidden Layer = 1, Epoch = 50) | 0.01 | 0.21 | 0.18 | 0.10 |
| Neural Network (Hidden Layer = 1, Epoch = 100) | 0.25 | 0.23 | 0.27 | 0.26 |
| Neural Network (Hidden Layer = 1, Epoch = 150) | 0.29 | 0.34 | 0.36 | 0.35 |
| Neural Network (Hidden Layer = 2, Epoch = 50) | 0.12 | 0.09 | 0.23 | 0.29 |
| Neural Network (Hidden Layer = 2, Epoch = 100) | 0.24 | 0.41 | 0.31 | 0.43 |
| Neural Network (Hidden Layer = 2, Epoch = 150) | 0.37 | 0.40 | 0.41 | 0.46 |
| Neural Network (Hidden Layer = 3, Epoch = 50) | 0.19 | 0.30 | 0.23 | 0.36 |
| Neural Network (Hidden Layer = 3, Epoch = 100) | 0.30 | 0.43 | 0.41 | 0.44 |
| Neural Network (Hidden Layer = 3, Epoch = 150) | 0.39 | 0.39 | 0.34 | 0.48 |

Moreover, from Table 5, we can see that the F1 score and MCC of different MDEM algorithms lie within [0.46–0.63] and [0.21–0.43] respectively. From Table 5, it is also evident that F1 score of MDEM in mean is the best amongst the all 19 algorithms considered, while its MCC is only next to that of LR, RF, SVM and one of the NNs.

To summarize, the performance of different MDEM algorithms varies between [83.65% − 95.82%] of the best algorithm under 2, 3, 5 and 7-Fold cross validation technique. Moreover, F1 score and MCC of different MDEM variants are within [0.46–0.63] and [0.21–0.43] respectively. As can be seen from Table 5, MDEM in mean has the highest F1 score, while its MCC is only next to that of LR, RF, SVM and 3 of the NNs.

As we have mentioned earlier, there are 719 records after the outliers and extreme values are removed from our PID dataset. Out of these 719 records under consideration, 477 records are non-diabetic and 242 are diabetic. So, the sample under consideration is not quite uniform. Rather, it is skewed to some extent towards non-diabetic records. A preferred choice to work with such imbalanced dataset is to use stratified k-fold cross validation technique instead of simple k-fold.

**Table 6**. Performance analyses of different algorithms using stratified k-fold cross validation technique for PID dataset.

| Algorithm | 2-Fold | 3-Fold | 5-Fold | 7-Fold |
|---|---|---|---|---|
| MDEM (Mean) | 73.99% | 73.02% | 73.85% | 73.85% |
| MDEM (Variance) | 70.79% | 70.93% | 70.78% | 70.65% |
| MDEM (3rd Central Moment) | 68.01% | 67.03% | 67.31% | 67.18% |
| MDEM (4th Central Moment) | 66.90% | 67.18% | 67.03% | 66.07% |
| Logistic Regression (LR) | 77.33% | 77.05% | 77.75% | 77.47% |
| Random Foresh (RF) | 75.66% | 76.78% | 77.33% | 79.42% |
| Support Vector Machine (SVM) | 76.50% | 76.36% | 77.61% | 77.48% |
| K Nearest Neighbor (KNN) (n = 3) | 72.60% | 73.99% | 73.30% | 73.99% |
| K Nearest Neighbor (KNN) (n = 5) | 73.58% | 73.72% | 74.27% | 74.28% |
| K Nearest Neighbor (KNN) (n = 7) | 74.55% | 73.58% | 73.57% | 73.30% |
| Neural Network (Hidden Layer = 1, Epoch = 50) | 66.62% | 66.76% | 67.32% | 69.40% |
| Neural Network (Hidden Layer = 1, Epoch = 100) | 68.70% | 73.16% | 70.65% | 71.36% |
| Neural Network (Hidden Layer = 1, Epoch = 150) | 70.10% | 73.58% | 74.13% | 74.83% |
| Neural Network (Hidden Layer = 2, Epoch = 50) | 71.63% | 71.22% | 68.15% | 69.68% |
| Neural Network (Hidden Layer = 2, Epoch = 100) | 67.87% | 74.97% | 74.13% | 73.45% |
| Neural Network (Hidden Layer = 2, Epoch = 150) | 72.32% | 75.66% | 75.52% | 76.51% |
| Neural Network (Hidden Layer = 3, Epoch = 50) | 69.40% | 68.15% | 71.20% | 73.43% |
| Neural Network (Hidden Layer = 3, Epoch = 100) | 74.13% | 73.72% | 74.55% | 75.24% |
| Neural Network (Hidden Layer = 3, Epoch = 150) | 76.08% | 76.78% | 74.41% | 76.50% |

In this part of this analysis, we are going to summarize the performances of different MDEM algorithms along with other state of the art algorithms under stratified 2, 3, 5 and 7-fold cross validation techniques. The detailed results are presented into Tables 6, 7 and 8, while the summarized statistics are shown graphically in Fig 3.

From Table 6, we can see that the accuracies of MDEM algorithms in mean, variance, 3rd and 4th central moment under stratified 2-fold cross validation are 73.99%, 70.79%, 68.01% and 66.90%. The best algorithm in this case is found to be Logistic Regression (LR) with a run time accuracy of 77.33%. So, MDEM algorithms run within the range of [86.51% − 95.68%] the best algorithm under consideration. This is graphically presented in Fig 3.

Additionally, from Table 8, we can see that the F1 score and MCC of different MDEM algorithms vary within the range of [0.46–0.64] and [0.23–0.44] respectively. It is evident from Table 8 that F1 score of MDEM in mean (0.64) is maximum amongst all the 19 algorithms considered and its MCC of 0.44 is only next to that of RF, LR and one of the NNs.

Under stratified 3-fold cross validation technique, accuracies of different MDEM algorithms are found to be 73.02%, 70.93%, 67.03% and 67.18%, where MDEM in mean is the best performing one, while MDEM in 3rd central moment is the worst performing one with run time accuracy of 73.02% and 67.03% respectively. The best algorithm under stratified 3-fold cross validation is Logistic Regression (LR) with an accuracy of 77.05%. So, as can be seen from Fig 3, different MDEM algorithms run within the range of [87.19% − 94.77%] of the best algorithm under consideration. Moreover, as can be seen from Table 8, F1 score and MCC of different MDEM algorithms vary within the range of [0.47–0.63] and [0.24–0.43] respectively. F1 score of MDEM in mean is the best amongst the 19 algorithms considered, while its MCC is only next to that of LR, RF, SVM and one of the NNs.

Moreover, the run time performances of different MDEM algorithms under stratified 5-fold cross validation technique are found to be 73.85%, 70.78%, 67.31% and 67.03%. The best algorithm under current scenario is Logistic Regression (LR) with an accuracy of 77.75%. This implies, MDEM algorithms run within the range of [86.21% − 94.98%] of the best algorithm under present consideration as can be seen from Fig 3. In addition, F1 score and MCC of the MDEM algorithms vary within [0.47–0.64] and [0.23–0.44] respectively (see Table 8). F1 score of MDEM in mean is the best amongst all the algorithms, while its MCC is next to that of SVM, LR, RF and one of the NNs.

Finally, we analyze the performance of different algorithms under stratified 7-fold cross validation technique. As can be seen from Table 4, MDEM in mean, variance, 3rd and 4th central moment have accuracies of 73.85%, 70.65%, 67.18%

**Table 7.** Precision and recall scores of different algorithms using stratified k-fold cross validation technique for PID dataset.

| Algorithm | 2-Fold | 3-Fold | 5-Fold | 7-Fold |
|---|---|---|---|---|
| Precision: | | | | |
| MDEM (Mean) | 0.59 | 0.58 | 0.60 | 0.60 |
| MDEM (Variance) | 0.56 | 0.56 | 0.56 | 0.56 |
| MDEM (3rd Central Moment) | 0.54 | 0.51 | 0.52 | 0.51 |
| MDEM (4th Central Moment) | 0.51 | 0.52 | 0.51 | 0.50 |
| Logistic Regression (LR) | 0.76 | 0.74 | 0.75 | 0.75 |
| Random Foresh (RF) | 0.70 | 0.71 | 0.71 | 0.74 |
| Support Vector Machine (SVM) | 0.73 | 0.71 | 0.74 | 0.75 |
| K Nearest Neighbor (KNN) (n = 3) | 0.60 | 0.63 | 0.62 | 0.64 |
| K Nearest Neighbor (KNN) (n = 5) | 0.63 | 0.64 | 0.65 | 0.64 |
| K Nearest Neighbor (KNN) (n = 7) | 0.66 | 0.64 | 0.64 | 0.63 |
| Neural Network (Hidden Layer = 1, Epoch = 50) | 0.60 | 0.68 | 0.61 | 0.48 |
| Neural Network (Hidden Layer = 1, Epoch = 100) | 0.64 | 0.46 | 0.54 | 0.67 |
| Neural Network (Hidden Layer = 1, Epoch = 150) | 0.60 | 0.64 | 0.70 | 0.70 |
| Neural Network (Hidden Layer = 2, Epoch = 50) | 0.59 | 0.47 | 0.69 | 0.57 |
| Neural Network (Hidden Layer = 2, Epoch = 100) | 0.67 | 0.68 | 0.74 | 0.69 |
| Neural Network (Hidden Layer = 2, Epoch = 150) | 0.69 | 0.69 | 0.73 | 0.71 |
| Neural Network (Hidden Layer = 3, Epoch = 50) | 0.64 | 0.41 | 0.68 | 0.67 |
| Neural Network (Hidden Layer = 3, Epoch = 100) | 0.67 | 0.67 | 0.66 | 0.68 |
| Neural Network (Hidden Layer = 3, Epoch = 150) | 0.69 | 0.66 | 0.59 | 0.57 |
| Recall: | | | | |
| MDEM (Mean) | 0.71 | 0.69 | 0.69 | 0.69 |
| MDEM (Variance) | 0.64 | 0.65 | 0.65 | 0.65 |
| MDEM (3rd Central Moment) | 0.38 | 0.37 | 0.40 | 0.40 |
| MDEM (4th Central Moment) | 0.42 | 0.43 | 0.44 | 0.49 |
| Logistic Regression (LR) | 0.47 | 0.50 | 0.51 | 0.51 |
| Random Foresh (RF) | 0.55 | 0.55 | 0.57 | 0.60 |
| Support Vector Machine (SVM) | 0.47 | 0.51 | 0.52 | 0.51 |
| K Nearest Neighbor (KNN) (n = 3) | 0.52 | 0.53 | 0.53 | 0.54 |
| K Nearest Neighbor (KNN) (n = 5) | 0.51 | 0.51 | 0.52 | 0.55 |
| K Nearest Neighbor (KNN) (n = 7) | 0.50 | 0.49 | 0.50 | 0.52 |
| Neural Network (Hidden Layer = 1, Epoch = 50) | 0.04 | 0.20 | 0.22 | 0.20 |
| Neural Network (Hidden Layer = 1, Epoch = 100) | 0.35 | 0.26 | 0.26 | 0.41 |
| Neural Network (Hidden Layer = 1, Epoch = 150) | 0.36 | 0.43 | 0.50 | 0.45 |
| Neural Network (Hidden Layer = 2, Epoch = 50) | 0.26 | 0.22 | 0.45 | 0.41 |
| Neural Network (Hidden Layer = 2, Epoch = 100) | 0.44 | 0.48 | 0.52 | 0.51 |
| Neural Network (Hidden Layer = 2, Epoch = 150) | 0.50 | 0.54 | 0.58 | 0.55 |
| Neural Network (Hidden Layer = 3, Epoch = 50) | 0.46 | 0.29 | 0.49 | 0.51 |
| Neural Network (Hidden Layer = 3, Epoch = 100) | 0.53 | 0.57 | 0.57 | 0.56 |
| Neural Network (Hidden Layer = 3, Epoch = 150) | 0.57 | 0.57 | 0.45 | 0.47 |

and 66.07% respectively. The best algorithm under stratified 7-fold cross validation is Random Forest (RF) with a run time accuracy of 79.42%. So, MDEM algorithms run within the range of [83.19% − 92.99%] of the best algorithm under consideration. Additionally, F1 score and MCC of different MDEM algorithms are found to be within [0.49–0.64] and [0.24–0.44] respectively. It is evident from Table 8 that MDEM in mean has the highest F1 score, while its MCC is only next to SVM, LR, RF and one of the NNs.

### 10.2 Wisconsin breast cancer dataset

Like the PID dataset, we use the same 19 algorithms on Wisconsin breast cancer database and compare their performances. The results obtained are presented in Tables 9–14. To be more precise, Table 9 contains the accuracy scores, Table 10 contains the precision and recall scores and Table 11 contains F1 scores and MCC of different algorithms under

**Table 8**. F1 scores and MCC of different algorithms using stratified k-fold cross validation technique for PID dataset.

| Algorithm | 2-Fold | 3-Fold | 5-Fold | 7-Fold |
|---|---|---|---|---|
| F1 Score: | | | | |
| MDEM (Mean) | 0.64 | 0.63 | 0.64 | 0.64 |
| MDEM (Variance) | 0.59 | 0.60 | 0.60 | 0.60 |
| MDEM (3rd Central Moment) | 0.45 | 0.43 | 0.45 | 0.45 |
| MDEM (4th Central Moment) | 0.46 | 0.47 | 0.47 | 0.49 |
| Logistic Regression (LR) | 0.58 | 0.59 | 0.61 | 0.60 |
| Random Foresh (RF) | 0.62 | 0.61 | 0.63 | 0.65 |
| Support Vector Machine (SVM) | 0.57 | 0.59 | 0.61 | 0.60 |
| K Nearest Neighbor (KNN) (n = 3) | 0.56 | 0.58 | 0.57 | 0.58 |
| K Nearest Neighbor (KNN) (n = 5) | 0.56 | 0.56 | 0.57 | 0.59 |
| K Nearest Neighbor (KNN) (n = 7) | 0.56 | 0.56 | 0.56 | 0.57 |
| Neural Network (Hidden Layer = 1, Epoch = 50) | 0.08 | 0.29 | 0.31 | 0.27 |
| Neural Network (Hidden Layer = 1, Epoch = 100) | 0.45 | 0.33 | 0.35 | 0.50 |
| Neural Network (Hidden Layer = 1, Epoch = 150) | 0.45 | 0.51 | 0.58 | 0.54 |
| Neural Network (Hidden Layer = 2, Epoch = 50) | 0.36 | 0.30 | 0.54 | 0.47 |
| Neural Network (Hidden Layer = 2, Epoch = 100) | 0.53 | 0.56 | 0.60 | 0.58 |
| Neural Network (Hidden Layer = 2, Epoch = 150) | 0.58 | 0.60 | 0.64 | 0.62 |
| Neural Network (Hidden Layer = 3, Epoch = 50) | 0.52 | 0.34 | 0.56 | 0.58 |
| Neural Network (Hidden Layer = 3, Epoch = 100) | 0.59 | 0.61 | 0.61 | 0.61 |
| Neural Network (Hidden Layer = 3, Epoch = 150) | 0.64 | 0.61 | 0.51 | 0.52 |
| MCC: | | | | |
| MDEM (Mean) | 0.44 | 0.43 | 0.44 | 0.44 |
| MDEM (Variance) | 0.37 | 0.38 | 0.38 | 0.38 |
| MDEM (3rd Central Moment) | 0.24 | 0.21 | 0.23 | 0.23 |
| MDEM (4th Central Moment) | 0.23 | 0.24 | 0.23 | 0.24 |
| Logistic Regression (LR) | 0.46 | 0.46 | 0.48 | 0.47 |
| Random Foresh (RF) | 0.46 | 0.45 | 0.48 | 0.51 |
| Support Vector Machine (SVM) | 0.44 | 0.45 | 0.47 | 0.47 |
| K Nearest Neighbor (KNN) (n = 3) | 0.36 | 0.40 | 0.38 | 0.40 |
| K Nearest Neighbor (KNN) (n = 5) | 0.38 | 0.39 | 0.40 | 0.41 |
| K Nearest Neighbor (KNN) (n = 7) | 0.40 | 0.38 | 0.38 | 0.38 |
| Neural Network (Hidden Layer = 1, Epoch = 50) | 0.07 | 0.22 | 0.21 | 0.19 |
| Neural Network (Hidden Layer = 1, Epoch = 100) | 0.30 | 0.24 | 0.25 | 0.35 |
| Neural Network (Hidden Layer = 1, Epoch = 150) | 0.28 | 0.34 | 0.43 | 0.40 |
| Neural Network (Hidden Layer = 2, Epoch = 50) | 0.22 | 0.22 | 0.40 | 0.33 |
| Neural Network (Hidden Layer = 2, Epoch = 100) | 0.38 | 0.39 | 0.44 | 0.43 |
| Neural Network (Hidden Layer = 2, Epoch = 150) | 0.43 | 0.45 | 0.50 | 0.47 |
| Neural Network (Hidden Layer = 3, Epoch = 50) | 0.36 | 0.22 | 0.40 | 0.41 |
| Neural Network (Hidden Layer = 3, Epoch = 100) | 0.42 | 0.44 | 0.44 | 0.45 |
| Neural Network (Hidden Layer = 3, Epoch = 150) | 0.47 | 0.44 | 0.40 | 0.37 |

k-fold cross validation technique. On the other hand, Table 12 documents the accuracy scores, Table 13 documents the precision and recall scores and Table 14 presents the F1 statistics and MCCs of all the 19 algorithms under stratified k-fold cross validation technique.

From Table 9, we can see that the accuracy scores of different MDEM algorithms vary between [86.77% − 93.00%] in 2-fold cross validation. The best algorithm in 2-fold is found to be a NN with 3 hidden layers and 100 epochs, which has an accuracy score of 97.47%. Thus, the MDEM algorithms runs within [89.02% − 95.41%] of the best algorithm in 2-fold cross validation technique. Moreover, from Table 11 we can see that the F1 scores and MCCs of different variants of MDEM fluctuate between [0.84–0.90] and [0.76–0.85]. The best F1 score and MCC are obtained for MDEM in mean (0.90 and 0.85), which is a bit lower than the best F1 score of 0.96 and 0.94 respectively.

Under 3-fold cross validation, MDEM algorithms have obtained accuracy scores within the range [86.77% − 93.97%]. The best amongst the MDEM algorithms is MDEM in mean with an accuracy score of 93.97%, while the overall best is

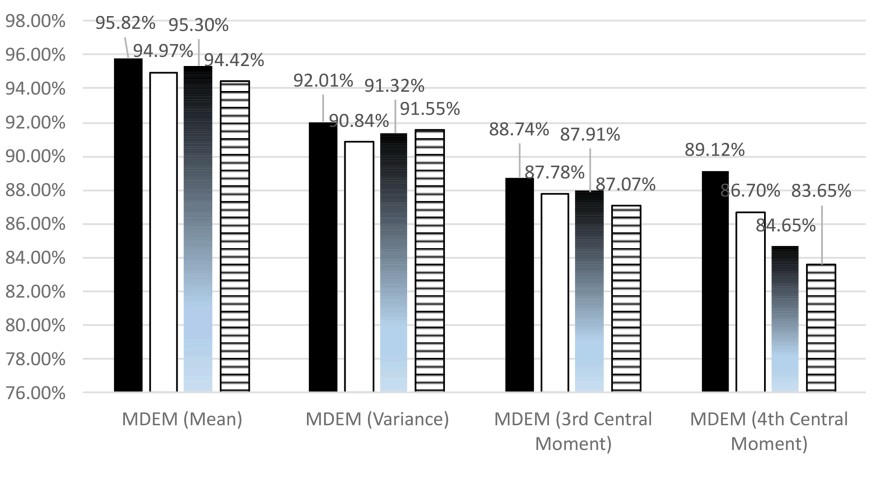

**Fig 2**. **Performance of MDEM algorithms as percentage of the best algorithms in k-fold for PID dataset.**

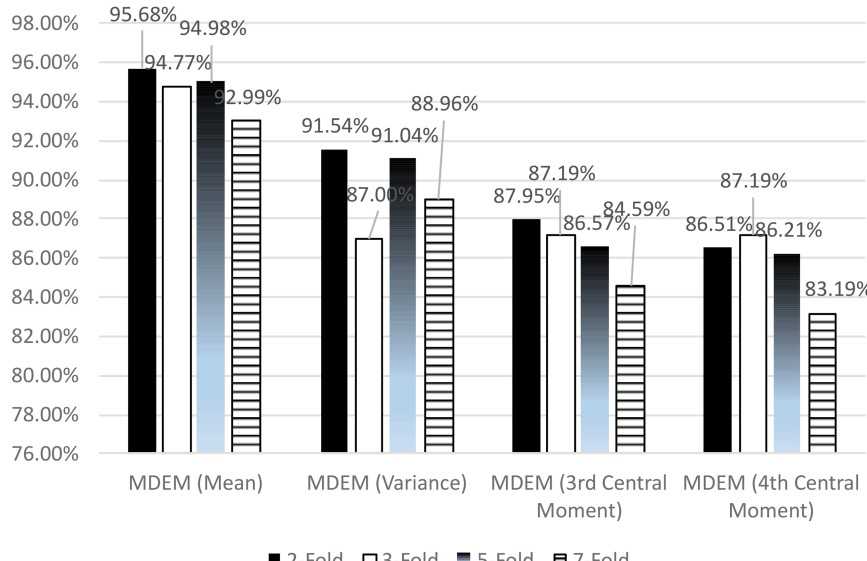

**Fig 3**. **Performance of MDEM algorithms as percentage of the best algorithms in stratified k-fold for PID dataset.**

97.66% (for NN with 3 hidden layers and 150 epochs). Thus, the performance of MDEM algorithms vary within the range of [88.85% − 96.22] of the best algorithm. Moreover, F1 score and MCC of the best MDEM algorithm are 0.91 and 0.86 respectively, which represent a good fit.

**Table 9**. Performance analyses of different algorithms using k-fold cross validation technique for Wisconsin breast cancer dataset.

| Algorithm | 2-Fold | 3-Fold | 5-Fold | 7-Fold |
|---|---|---|---|---|
| MDEM (Mean) | 93.00% | 93.97% | 93.79% | 93.96% |
| MDEM (Variance) | 91.25% | 92.60% | 92.02% | 92.78% |
| MDEM (3rd Central Moment) | 86.77% | 86.77% | 87.74% | 88.71% |
| MDEM (4th Central Moment) | 88.33% | 89.49% | 89.49% | 89.28% |
| Logistic Regression (LR) | 95.53% | 96.11% | 96.70% | 96.30% |
| Random Foresh (RF) | 94.36% | 96.30% | 95.53% | 96.30% |
| Support Vector Machine (SVM) | 97.08% | 97.08% | 97.28% | 97.67% |
| K Nearest Neighbor (KNN) (n = 3) | 95.72% | 95.91% | 96.30% | 96.50% |
| K Nearest Neighbor (KNN) (n = 5) | 95.53% | 95.91% | 96.11% | 96.30% |
| K Nearest Neighbor (KNN) (n = 7) | 95.91% | 96.30% | 96.11% | 96.30% |
| Neural Network (Hidden Layer = 1, Epoch = 50) | 91.05% | 91.43% | 93.19% | 92.41% |
| Neural Network (Hidden Layer = 1, Epoch = 100) | 93.97% | 95.33% | 95.53% | 95.33% |
| Neural Network (Hidden Layer = 1, Epoch = 150) | 94.94% | 94.74% | 96.50% | 96.10% |
| Neural Network (Hidden Layer = 2, Epoch = 50) | 90.47% | 94.94% | 95.92% | 95.33% |
| Neural Network (Hidden Layer = 2, Epoch = 100) | 95.91% | 95.91% | 96.89% | 97.47% |
| Neural Network (Hidden Layer = 2, Epoch = 150) | 96.11% | 97.47% | 97.09% | 97.47% |
| Neural Network (Hidden Layer = 3, Epoch = 50) | 90.66% | 95.13% | 90.10% | 95.91% |
| Neural Network (Hidden Layer = 3, Epoch = 100) | 95.91% | 97.66% | 96.70% | 97.47% |
| Neural Network (Hidden Layer = 3, Epoch = 150) | 97.47% | 97.66% | 96.69% | 97.27% |

For 5-fold cross validation, the accuracy scores of MDEM algorithms are found to be within [87.74% − 93.79%], while the best accuracy score of 97.09% is obtained for a NN with 2 hidden layers and 150 epochs (Table 9). Thus, the accuracy scores of MDEM algorithms vary within [90.47% − 96.60%] of the best algorithm under consideration. Additionally, the F1 score and MCC of the best MDEM algorithm is found to be 0.90 and 0.86 respectively, which also represent a good fit.

Next, the accuracy scores of the MDEM algorithms under 7-fold cross validation are fond to fluctuate within [88.71% − 93.96%], while the best accuracy score of 97.67% is obtained for SVM. Thus, the accuracy scores of MDEM algorithms vary within the [90.83% − 96.20%] range of the best algorithm under consideration. Additionally, the F1 score and MCC of the best MDEM algorithm turn out to to be 0.91 and 0.86 respectively, which are obtained for MDEM in mean. As can be seen from Table 11, the best F1 score of 0.97 and MCC of 0.95 are obtained for SVM, which are quite comparable to those of MDEM in mean.

Accuracy scores of different MDEM algorithms as percentage of the performance metric of the best algorithm under consideration in k-fold cross validation are graphically presented in Fig 4.

As we are done with the analysis of our algorithms under k-fold cross validation technique, we now compare their performances using stratified k-fold cross validation. To start with, the accuracy scores, precision and recall scores along with the corresponding F1 scores and MCCs are presented in Tables 12, 13 and 14 respectively.

From Table 12, we can see that accuracy scores of different MDEM algorithms vary between [86.77% − 93.77%] in 2-fold stratified cross validation. The best algorithm in 2-fold is found to be a NN with 3 hidden layers and 100 epochs with an accuracy score 97.28%. Thus, the performance of the MDEM algorithms are within the range of [89.20% − 96.40%] of the best algorithm under consideration. F1 score and MCC of the best MDEM algorithm are found to be 0.90 and 0.86 respectively (Table 14).

For stratified 3-fold cross validation, the best algorithm is found to be a NN with 3 hidden layers and 100 epochs yielding an accuracy score of 98.25%. The accuracy scores of different MDEM algorithms are noted to be within [87.55% − 93.97%], which are within [89.11% − 95.64%] range of the best algorithm. F1 score and MCC of the best MDEM algorithm are found to be 0.91 and 0.87 respectively (Table 14).

For stratified 5-fold cross validation, the best algorithm is a NN with 3 hidden layers and 100 epochs, having an accuracy score of 97.47%, while the performance of MDEM algorithms vary between [87.94% − 94.16%]. So, MDEMs run

**Table 10**. Precision and recall scores of different algorithms using k-fold cross validation technique for Wisconsin breast cancer dataset.

| Algorithm | 2-Fold | 3-Fold | 5-Fold | 7-Fold |
|---|---|---|---|---|
| Precision: | | | | |
| MDEM (Mean) | 0.94 | 0.94 | 0.94 | 0.95 |
| MDEM (Variance) | 0.83 | 0.85 | 0.85 | 0.86 |
| MDEM (3rd Central Moment) | 0.77 | 0.77 | 0.78 | 0.80 |
| MDEM (4th Central Moment) | 0.79 | 0.81 | 0.81 | 0.81 |
| Logistic Regression (LR) | 0.99 | 1.00 | 1.00 | 0.99 |
| Random Foresh (RF) | 0.92 | 0.95 | 0.94 | 0.96 |
| Support Vector Machine (SVM) | 0.99 | 0.99 | 0.99 | 0.99 |
| K Nearest Neighbor (KNN) (n = 3) | 0.97 | 0.97 | 0.97 | 0.98 |
| K Nearest Neighbor (KNN) (n = 5) | 0.98 | 0.98 | 0.98 | 0.98 |
| K Nearest Neighbor (KNN) (n = 7) | 0.98 | 0.98 | 0.98 | 0.98 |
| Neural Network (Hidden Layer = 1, Epoch = 50) | 0.9 | 0.96 | 0.94 | 0.90 |
| Neural Network (Hidden Layer = 1, Epoch = 100) | 0.96 | 0.97 | 0.99 | 0.96 |
| Neural Network (Hidden Layer = 1, Epoch = 150) | 0.96 | 0.96 | 0.98 | 0.96 |
| Neural Network (Hidden Layer = 2, Epoch = 50) | 0.95 | 0.95 | 0.96 | 0.94 |
| Neural Network (Hidden Layer = 2, Epoch = 100) | 0.95 | 0.97 | 0.97 | 0.98 |
| Neural Network (Hidden Layer = 2, Epoch = 150) | 0.95 | 0.98 | 0.98 | 0.98 |
| Neural Network (Hidden Layer = 3, Epoch = 50) | 0.94 | 0.97 | 0.78 | 0.97 |
| Neural Network (Hidden Layer = 3, Epoch = 100) | 0.96 | 0.98 | 0.98 | 0.98 |
| Neural Network (Hidden Layer = 3, Epoch = 150) | 0.97 | 0.99 | 0.96 | 0.97 |
| Recall: | | | | |
| MDEM (Mean) | 0.86 | 0.87 | 0.87 | 0.87 |
| MDEM (Variance) | 0.93 | 0.95 | 0.92 | 0.93 |
| MDEM (3rd Central Moment) | 0.86 | 0.87 | 0.88 | 0.89 |
| MDEM (4th Central Moment) | 0.9 | 0.90 | 0.90 | 0.90 |
| Logistic Regression (LR) | 0.87 | 0.88 | 0.90 | 0.90 |
| Random Foresh (RF) | 0.9 | 0.93 | 0.93 | 0.93 |
| Support Vector Machine (SVM) | 0.92 | 0.92 | 0.93 | 0.93 |
| K Nearest Neighbor (KNN) (n = 3) | 0.9 | 0.90 | 0.92 | 0.92 |
| K Nearest Neighbor (KNN) (n = 5) | 0.89 | 0.90 | 0.92 | 0.92 |
| K Nearest Neighbor (KNN) (n = 7) | 0.89 | 0.91 | 0.91 | 0.91 |
| Neural Network (Hidden Layer = 1, Epoch = 50) | 0.83 | 0.79 | 0.85 | 0.88 |
| Neural Network (Hidden Layer = 1, Epoch = 100) | 0.85 | 0.89 | 0.88 | 0.90 |
| Neural Network (Hidden Layer = 1, Epoch = 150) | 0.89 | 0.88 | 0.91 | 0.92 |
| Neural Network (Hidden Layer = 2, Epoch = 50) | 0.77 | 0.90 | 0.92 | 0.93 |
| Neural Network (Hidden Layer = 2, Epoch = 100) | 0.92 | 0.91 | 0.92 | 0.94 |
| Neural Network (Hidden Layer = 2, Epoch = 150) | 0.93 | 0.94 | 0.94 | 0.94 |
| Neural Network (Hidden Layer = 3, Epoch = 50) | 0.86 | 0.89 | 0.73 | 0.91 |
| Neural Network (Hidden Layer = 3, Epoch = 100) | 0.92 | 0.95 | 0.92 | 0.95 |
| Neural Network (Hidden Layer = 3, Epoch = 150) | 0.95 | 0.94 | 0.94 | 0.95 |

within the [90.22% − 96.60%] range of the best algorithm. F1 score and MCC of the best MDEM algorithm are found to be 0.91 and 0.87 respectively (Table 14).

Finally, for 7-fold stratified cross validation, the best algorithm is a NN with an accuracy score of 97.86% (NN with 3 hidden layers and 100 epochs). On the other hand, the accuracy scores of MDEM algorithms vary within [87.95% − 94.17%], which is very close to the accuracy scores of the best algorithm. F1 score and MCC of the best MDEM algorithm are found to be 0.91 and 0.87 respectively, which represent a quite good fit (Table 14).

Accuracy scores of different MDEM algorithms as percentage of the performance metric of the best algorithm under consideration in stratified k-fold cross validation are graphically presented in Fig 5.

**Table 11. F1 scores and MCCs of different algorithms using k-fold cross validation technique for Wisconsin breast cancer dataset.**

| Algorithm | 2-Fold | 3-Fold | 5-Fold | 7-Fold |
|---|---|---|---|---|
| F1 Score: | | | | |
| MDEM (Mean) | 0.90 | 0.91 | 0.90 | 0.91 |
| MDEM (Variance) | 0.88 | 0.90 | 0.89 | 0.90 |
| MDEM (3rd Central Moment) | 0.82 | 0.82 | 0.83 | 0.84 |
| MDEM (4th Central Moment) | 0.84 | 0.85 | 0.85 | 0.85 |
| Logistic Regression (LR) | 0.93 | 0.94 | 0.95 | 0.94 |
| Random Foresh (RF) | 0.92 | 0.94 | 0.93 | 0.94 |
| Support Vector Machine (SVM) | 0.96 | 0.96 | 0.96 | 0.97 |
| K Nearest Neighbor (KNN) (n = 3) | 0.93 | 0.93 | 0.94 | 0.95 |
| K Nearest Neighbor (KNN) (n = 5) | 0.93 | 0.94 | 0.94 | 0.94 |
| K Nearest Neighbor (KNN) (n = 7) | 0.93 | 0.94 | 0.94 | 0.94 |
| Neural Network (Hidden Layer = 1, Epoch = 50) | 0.86 | 0.85 | 0.89 | 0.89 |
| Neural Network (Hidden Layer = 1, Epoch = 100) | 0.90 | 0.93 | 0.93 | 0.93 |
| Neural Network (Hidden Layer = 1, Epoch = 150) | 0.92 | 0.91 | 0.95 | 0.94 |
| Neural Network (Hidden Layer = 2, Epoch = 50) | 0.84 | 0.92 | 0.94 | 0.93 |
| Neural Network (Hidden Layer = 2, Epoch = 100) | 0.93 | 0.94 | 0.95 | 0.96 |
| Neural Network (Hidden Layer = 2, Epoch = 150) | 0.94 | 0.96 | 0.96 | 0.96 |
| Neural Network (Hidden Layer = 3, Epoch = 50) | 0.90 | 0.92 | 0.75 | 0.94 |
| Neural Network (Hidden Layer = 3, Epoch = 100) | 0.93 | 0.97 | 0.95 | 0.96 |
| Neural Network (Hidden Layer = 3, Epoch = 150) | 0.96 | 0.97 | 0.95 | 0.96 |
| MCC: | | | | |
| MDEM (Mean) | 0.85 | 0.86 | 0.86 | 0.86 |
| MDEM (Variance) | 0.81 | 0.84 | 0.83 | 0.84 |
| MDEM (3rd Central Moment) | 0.72 | 0.72 | 0.74 | 0.77 |
| MDEM (4th Central Moment) | 0.76 | 0.78 | 0.77 | 0.77 |
| Logistic Regression (LR) | 0.90 | 0.91 | 0.93 | 0.92 |
| Random Foresh (RF) | 0.87 | 0.92 | 0.90 | 0.92 |
| Support Vector Machine (SVM) | 0.94 | 0.93 | 0.94 | 0.95 |
| K Nearest Neighbor (KNN) (n = 3) | 0.90 | 0.90 | 0.92 | 0.92 |
| K Nearest Neighbor (KNN) (n = 5) | 0.90 | 0.91 | 0.91 | 0.92 |
| K Nearest Neighbor (KNN) (n = 7) | 0.91 | 0.92 | 0.91 | 0.92 |
| Neural Network (Hidden Layer = 1, Epoch = 50) | 0.80 | 0.81 | 0.85 | 0.83 |
| Neural Network (Hidden Layer = 1, Epoch = 100) | 0.86 | 0.89 | 0.90 | 0.90 |
| Neural Network (Hidden Layer = 1, Epoch = 150) | 0.88 | 0.88 | 0.92 | 0.91 |
| Neural Network (Hidden Layer = 2, Epoch = 50) | 0.79 | 0.89 | 0.91 | 0.90 |
| Neural Network (Hidden Layer = 2, Epoch = 100) | 0.90 | 0.91 | 0.93 | 0.94 |
| Neural Network (Hidden Layer = 2, Epoch = 150) | 0.91 | 0.94 | 0.94 | 0.94 |
| Neural Network (Hidden Layer = 3, Epoch = 50) | 0.86 | 0.89 | 0.73 | 0.91 |
| Neural Network (Hidden Layer = 3, Epoch = 100) | 0.91 | 0.95 | 0.93 | 0.94 |
| Neural Network (Hidden Layer = 3, Epoch = 150) | 0.94 | 0.95 | 0.92 | 0.94 |

## 10.3 Run time analysis

In this subsection, we will analyze the running time of different algorithms using k-fold cross validation techniques. Although not reported here, stratified k-fold cross validation technique also entails equivalent results. Results using k-fold cross validation are tabulated in Tables 15–18. To be precise, Tables 15 and 16 summarize the running time for Pima Indian Diabetes dataset, while Tables 17 and 18 summarize the same for Wisconsin breast cancer dataset. We will discuss the running time analysis of different algorithms using PID dataset and Wisconsin breast cancer dataset in separate sections.

**Table 12. Performance analyses of different algorithms using stratified k-fold cross validation technique for Wisconsin breast cancer dataset.**

| Algorithm | 2-Fold | 3-Fold | 5-Fold | 7-Fold |
|---|---|---|---|---|
| MDEM (Mean) | 93.77% | 93.97% | 94.16% | 94.17% |
| MDEM (Variance) | 91.44% | 91.44% | 92.22% | 91.83% |
| MDEM (3rd Central Moment) | 86.77% | 87.55% | 87.94% | 87.95% |
| MDEM (4th Central Moment) | 89.30% | 89.49% | 90.08% | 90.08% |
| Logistic Regression (LR) | 94.74% | 95.91% | 96.69% | 96.11% |
| Random Foresh (RF) | 95.53% | 95.53% | 95.92% | 95.73% |
| Support Vector Machine (SVM) | 96.69% | 97.28% | 98.06% | 97.65% |
| K Nearest Neighbor (KNN) (n = 3) | 96.88% | 96.69% | 96.91% | 96.49% |
| K Nearest Neighbor (KNN) (n = 5) | 95.91% | 96.31% | 96.50% | 96.50% |
| K Nearest Neighbor (KNN) (n = 7) | 96.30% | 96.89% | 96.30% | 96.31% |
| Neural Network (Hidden Layer = 1, Epoch = 50) | 89.88% | 93.58% | 93.58% | 93.97% |
| Neural Network (Hidden Layer = 1, Epoch = 100) | 93.77% | 93.94% | 95.33% | 95.32% |
| Neural Network (Hidden Layer = 1, Epoch = 150) | 95.93% | 95.72% | 96.50% | 95.72% |
| Neural Network (Hidden Layer = 2, Epoch = 50) | 94.16% | 92.80% | 95.14% | 96.50% |
| Neural Network (Hidden Layer = 2, Epoch = 100) | 96.30% | 96.70% | 96.70% | 97.67% |
| Neural Network (Hidden Layer = 2, Epoch = 150) | 96.69% | 96.89% | 97.28% | 97.28% |
| Neural Network (Hidden Layer = 3, Epoch = 50) | 94.75% | 95.72% | 96.89% | 95.91% |
| Neural Network (Hidden Layer = 3, Epoch = 100) | 97.28% | 98.25% | 97.08% | 97.86% |
| Neural Network (Hidden Layer = 3, Epoch = 150) | 96.69% | 97.47% | 97.47% | 97.09% |

**10.3.1 Running time analysis using PID dataset.** From Table 15, we can see that the training time of MDEM in mean under 2, 3, 5 and 7-fold cross validation technique using PID dataset is best amongst all the simulated MDEM algorithms, which nicely converges with our previous theoretical analysis. Moreover, MDEM in mean has a significantly lower running time in training phase as compared to LR, RF, SVM and all the nine NN variants. However, all the 03 KNN run better than MDEM in mean, as we have discussed earlier that KNNs have constant running time in the training phase as almost zero pre-processing is needed at this stage. But, KNNs performance suffer much in the testing phase as its time complexity is linear on all the training and testing data. This is evident from Table 16 that MDEM in mean runs noticeably well in the testing phase as compared to all the KNN variants. In testing phase, MDEM in mean runs better than all the 03 KNNs as well as all the nine NN variants as can be seen from Table 16. However, it is outperformed by LR, RF and SVM in the testing phase.

**10.3.2 Running time analysis using Wisconsin breast cancer dataset.** For Wisconsin breast cancer dataset also, the best MDEM algorithm in training and testing phase is again MDEM in mean, as can be seen from Tables 17 and 18. From Table 17, we can see that MDEM in mean runs better than LR, RF and all the nine NN variants in the training phase. But, it is outperformed by SVM and all the 03 KNN variants. On the other hand, in testing phase, MDEM in mean runs better than all the 03 KNN and 09 NN variants. However, it is outperformed by LR, RF and SVM.

## 10.4 Discussion

From the above discussion, it can be seen that the MDEM algorithm performs within the $[83.19\% - 95.82\%]$ bound of the best algorithm under consideration for PID dataset. Moreover, for Wisconsin breast cancer dataset, its performance seems to swing between $[88.85\% - 96.41\%]$. Moreover, for both the datasets, MDEMs involving lower order moments perform remarkably well as compared to MDEMs involving higher order moments. In fact, the lower bounds of the aforementioned accuracy scores are attributed to MDEMs involving 3rd and 4th order central moments, while the upper bounds

**Table 13**. Precision and recall scores of different algorithms using stratified k-fold cross validation technique for Wisconsin breast cancer dataset.

| Algorithm | 2-Fold | 3-Fold | 5-Fold | 7-Fold |
|---|---|---|---|---|
| Precision: | | | | |
| MDEM (Mean) | 0.96 | 0.96 | 0.96 | 0.96 |
| MDEM (Variance) | 0.85 | 0.85 | 0.86 | 0.87 |
| MDEM (3rd Central Moment) | 0.77 | 0.79 | 0.79 | 0.80 |
| MDEM (4th Central Moment) | 0.81 | 0.81 | 0.83 | 0.83 |
| Logistic Regression (LR) | 0.99 | 0.99 | 0.99 | 0.99 |
| Random Foresh (RF) | 0.94 | 0.95 | 0.95 | 0.96 |
| Support Vector Machine (SVM) | 0.98 | 0.98 | 0.99 | 0.99 |
| K Nearest Neighbor (KNN) (n = 3) | 0.98 | 0.98 | 0.98 | 0.98 |
| K Nearest Neighbor (KNN) (n = 5) | 0.98 | 0.98 | 0.97 | 0.98 |
| K Nearest Neighbor (KNN) (n = 7) | 0.98 | 0.99 | 0.98 | 0.99 |
| Neural Network (Hidden Layer = 1, Epoch = 50) | 0.90 | 0.90 | 0.93 | 0.94 |
| Neural Network (Hidden Layer = 1, Epoch = 100) | 0.94 | 0.96 | 0.96 | 0.98 |
| Neural Network (Hidden Layer = 1, Epoch = 150) | 0.97 | 0.96 | 0.97 | 0.97 |
| Neural Network (Hidden Layer = 2, Epoch = 50) | 0.91 | 0.93 | 0.96 | 0.98 |
| Neural Network (Hidden Layer = 2, Epoch = 100) | 0.95 | 0.96 | 0.97 | 0.97 |
| Neural Network (Hidden Layer = 2, Epoch = 150) | 0.98 | 0.97 | 0.97 | 0.97 |
| Neural Network (Hidden Layer = 3, Epoch = 50) | 0.95 | 0.94 | 0.97 | 0.96 |
| Neural Network (Hidden Layer = 3, Epoch = 100) | 0.98 | 0.98 | 0.97 | 0.98 |
| Neural Network (Hidden Layer = 3, Epoch = 150) | 0.96 | 0.98 | 0.98 | 0.97 |
| Recall: | | | | |
| MDEM (Mean) | 0.86 | 0.86 | 0.87 | 0.87 |
| MDEM (Variance) | 0.91 | 0.91 | 0.92 | 0.91 |
| MDEM (3rd Central Moment) | 0.87 | 0.88 | 0.88 | 0.88 |
| MDEM (4th Central Moment) | 0.90 | 0.91 | 0.90 | 0.91 |
| Logistic Regression (LR) | 0.86 | 0.89 | 0.91 | 0.89 |
| Random Foresh (RF) | 0.93 | 0.91 | 0.93 | 0.91 |
| Support Vector Machine (SVM) | 0.92 | 0.94 | 0.95 | 0.94 |
| K Nearest Neighbor (KNN) (n = 3) | 0.93 | 0.92 | 0.91 | 0.91 |
| K Nearest Neighbor (KNN) (n = 5) | 0.90 | 0.91 | 0.93 | 0.91 |
| K Nearest Neighbor (KNN) (n = 7) | 0.90 | 0.92 | 0.91 | 0.90 |
| Neural Network (Hidden Layer = 1, Epoch = 50) | 0.79 | 0.91 | 0.87 | 0.88 |
| Neural Network (Hidden Layer = 1, Epoch = 100) | 0.87 | 0.89 | 0.90 | 0.89 |
| Neural Network (Hidden Layer = 1, Epoch = 150) | 0.90 | 0.91 | 0.93 | 0.90 |
| Neural Network (Hidden Layer = 2, Epoch = 50) | 0.93 | 0.86 | 0.89 | 0.92 |
| Neural Network (Hidden Layer = 2, Epoch = 100) | 0.94 | 0.94 | 0.93 | 0.96 |
| Neural Network (Hidden Layer = 2, Epoch = 150) | 0.93 | 0.94 | 0.95 | 0.95 |
| Neural Network (Hidden Layer = 3, Epoch = 50) | 0.89 | 0.93 | 0.94 | 0.92 |
| Neural Network (Hidden Layer = 3, Epoch = 100) | 0.94 | 0.97 | 0.94 | 0.96 |
| Neural Network (Hidden Layer = 3, Epoch = 150) | 0.94 | 0.95 | 0.95 | 0.95 |

are practically due to MDEMs in mean. Also, as we can be seen from Tables 5, 8, 11 and 14, MDEMs in mean and variance have consistently obtained better F1 score and MCC as compared to MDEMs in skewness and kurtosis. However, which version of MDEM algorithm will perform better than the others depends mostly on the underlying dataset used in the analysis. If the dataset favors lower order moments like mean and variance over higher order ones like skewness and kurtosis, then using lower order moments delivers better performance and, in our cases, we have reported exactly the same incident. However, there may be other datasets that prefer homogeneity in skewness and kurtosis to that in mean and variance and, for those datasets, MDEMs involving higher order moments may perform better.

**Table 14. F1 scores and MCCs of different algorithms using stratified k-fold cross validation technique for Wisconsin breast cancer dataset.**

| Algorithm | 2-Fold | 3-Fold | 5-Fold | 0.00 |
|---|---|---|---|---|
| F1 Score: | | | | |
| MDEM (Mean) | 0.90 | 0.91 | 0.91 | 0.91 |
| MDEM (Variance) | 0.88 | 0.88 | 0.89 | 0.88 |
| MDEM (3rd Central Moment) | 0.82 | 0.83 | 0.83 | 0.83 |
| MDEM (4th Central Moment) | 0.85 | 0.85 | 0.86 | 0.86 |
| Logistic Regression (LR) | 0.92 | 0.94 | 0.95 | 0.94 |
| Random Foresh (RF) | 0.93 | 0.93 | 0.94 | 0.93 |
| Support Vector Machine (SVM) | 0.95 | 0.96 | 0.97 | 0.96 |
| K Nearest Neighbor (KNN) (n = 3) | 0.95 | 0.95 | 0.94 | 0.95 |
| K Nearest Neighbor (KNN) (n = 5) | 0.94 | 0.94 | 0.95 | 0.95 |
| K Nearest Neighbor (KNN) (n = 7) | 0.94 | 0.95 | 0.94 | 0.94 |
| Neural Network (Hidden Layer = 1, Epoch = 50) | 0.84 | 0.91 | 0.90 | 0.90 |
| Neural Network (Hidden Layer = 1, Epoch = 100) | 0.90 | 0.92 | 0.93 | 0.93 |
| Neural Network (Hidden Layer = 1, Epoch = 150) | 0.93 | 0.94 | 0.95 | 0.93 |
| Neural Network (Hidden Layer = 2, Epoch = 50) | 0.91 | 0.89 | 0.92 | 0.95 |
| Neural Network (Hidden Layer = 2, Epoch = 100) | 0.94 | 0.95 | 0.95 | 0.97 |
| Neural Network (Hidden Layer = 2, Epoch = 150) | 0.95 | 0.95 | 0.96 | 0.96 |
| Neural Network (Hidden Layer = 3, Epoch = 50) | 0.92 | 0.94 | 0.95 | 0.94 |
| Neural Network (Hidden Layer = 3, Epoch = 100) | 0.96 | 0.97 | 0.96 | 0.97 |
| Neural Network (Hidden Layer = 3, Epoch = 150) | 0.95 | 0.96 | 0.96 | 0.96 |
| MCC: | | | | |
| MDEM (Mean) | 0.86 | 0.87 | 0.87 | 0.87 |
| MDEM (Variance) | 0.81 | 0.81 | 0.83 | 0.82 |
| MDEM (3rd Central Moment) | 0.72 | 0.74 | 0.74 | 0.74 |
| MDEM (4th Central Moment) | 0.77 | 0.79 | 0.79 | 0.79 |
| Logistic Regression (LR) | 0.88 | 0.91 | 0.93 | 0.91 |
| Random Foresh (RF) | 0.90 | 0.90 | 0.91 | 0.90 |
| Support Vector Machine (SVM) | 0.93 | 0.94 | 0.96 | 0.95 |
| K Nearest Neighbor (KNN) (n = 3) | 0.93 | 0.93 | 0.91 | 0.92 |
| K Nearest Neighbor (KNN) (n = 5) | 0.91 | 0.92 | 0.92 | 0.92 |
| K Nearest Neighbor (KNN) (n = 7) | 0.91 | 0.93 | 0.92 | 0.92 |
| Neural Network (Hidden Layer = 1, Epoch = 50) | 0.77 | 0.86 | 0.86 | 0.86 |
| Neural Network (Hidden Layer = 1, Epoch = 100) | 0.86 | 0.89 | 0.90 | 0.90 |
| Neural Network (Hidden Layer = 1, Epoch = 150) | 0.90 | 0.90 | 0.92 | 0.90 |
| Neural Network (Hidden Layer = 2, Epoch = 50) | 0.87 | 0.84 | 0.89 | 0.92 |
| Neural Network (Hidden Layer = 2, Epoch = 100) | 0.92 | 0.93 | 0.93 | 0.95 |
| Neural Network (Hidden Layer = 2, Epoch = 150) | 0.93 | 0.93 | 0.94 | 0.94 |
| Neural Network (Hidden Layer = 3, Epoch = 50) | 0.88 | 0.90 | 0.93 | 0.91 |
| Neural Network (Hidden Layer = 3, Epoch = 100) | 0.94 | 0.96 | 0.93 | 0.95 |
| Neural Network (Hidden Layer = 3, Epoch = 150) | 0.93 | 0.94 | 0.94 | 0.94 |

MDEM algorithms are better suited in numerous situations, where speed in running time is more important than cutting-edge accuracy. For example, MDEM algorithms can be practically deployed to different text classification scenarios, e.g., classifying documents according to search query, email filtering (spam vs non-spam and also categorization and labelling of emails), categorization of news articles (sports, politics, international etc.), populating newsfeeds for users in social media, sentiment analysis (classifying reviews as positive, negative and neutral), support ticket routing (assign incoming tickets to the right department), legal and medical documents tagging (tag documents based on predefined categories) and the alike. These are some of the applications, where speed is preferred to optimality as the data here are massive in quantity and consequences of possible misclassification are relatively less dangerous.

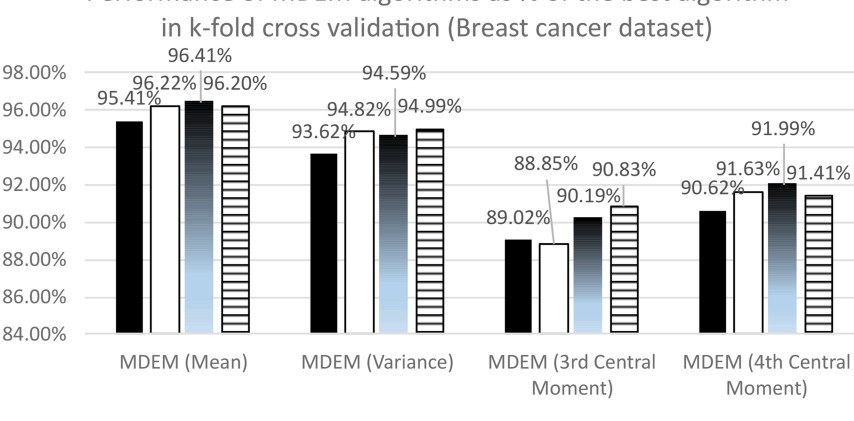

**Fig 4**. **Performance of MDEM algorithms as percentage of the best algorithms in k-fold for Wisconsin breast cancer dataset.**

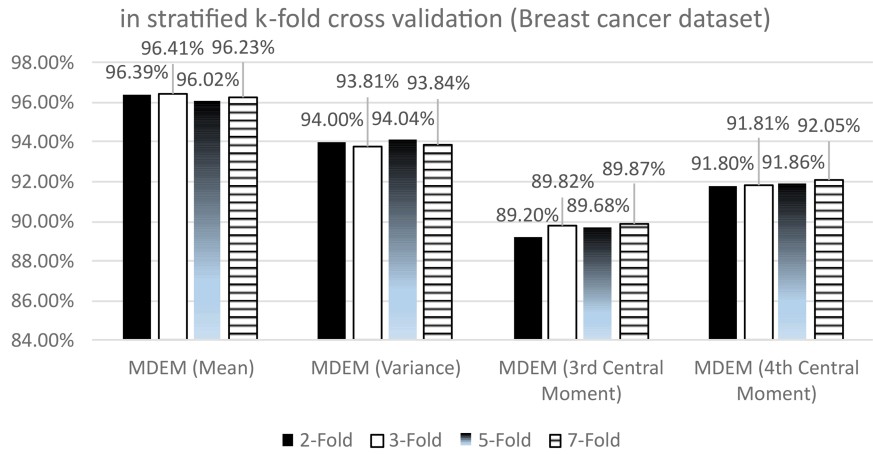

**Fig 5**. **Performance of MDEM algorithms as percentage of the best algorithms in stratified k-fold for Wisconsin breast cancer dataset.**

## 11 Conclusion

Here, we have proposed a new supervised learning algorithm that can train itself in times linear to the number of training rows. After the training phase, the algorithm can effectively classify every new instance in constant time for a particular problem and can instantly change its definition after each such inclusion. As we have discussed throughout this article, this significant improvement in running time is obtained at the cost of a slightly less than optimal performance. For the Pima Indian Diabetes (PID) dataset, our algorithms are found to perform within the range of [83.19% − 95.82%] of the best algorithm at hand. For Wisconsin breast cancer dataset, our algorithms perform within the range of [88.85 − 96.41]% of the best algorithm under consideration. So, whenever we are in need of a multiclass learning algorithm that is intended to

**Table 15**. Training time analysis of different algorithms for PID dataset using k-fold cross validation technique.

| Algorithm | Avg. training time per fold (in second) | | | |
|---|---|---|---|---|
| | 2-Fold | 3-Fold | 5-Fold | 7-Fold |
| MDEM (Mean) | 0.00076 | 0.00099 | 0.00149 | 0.00134 |
| MDEM (Variance) | 0.00156 | 0.00207 | 0.00250 | 0.00268 |
| MDEM (3rd Central Moment) | 0.00231 | 0.00306 | 0.00379 | 0.00405 |
| MDEM (4th Central Moment) | 0.00312 | 0.00414 | 0.00517 | 0.00545 |
| Logistic Regression (LR) | 0.00148 | 0.00151 | 0.00274 | 0.00141 |
| Random Foresh (RF) | 0.08689 | 0.09403 | 0.15115 | 0.10444 |
| Support Vector Machine (SVM) | 0.00173 | 0.00246 | 0.00505 | 0.00361 |
| K Nearest Neighbor (KNN) (n = 3) | 0.00050 | 0.00054 | 0.00133 | 0.00068 |
| K Nearest Neighbor (KNN) (n = 5) | 0.00049 | 0.00063 | 0.00064 | 0.00069 |
| K Nearest Neighbor (KNN) (n = 7) | 0.00056 | 0.00056 | 0.00063 | 0.00120 |
| Neural Network (Hidden Layer = 1, Epoch = 50) | 2.67306 | 2.92841 | 3.07796 | 3.28022 |
| Neural Network (Hidden Layer = 1, Epoch = 100) | 4.87762 | 5.27202 | 5.78326 | 8.71747 |
| Neural Network (Hidden Layer = 1, Epoch = 150) | 7.07480 | 7.81752 | 8.51187 | 8.62168 |
| Neural Network (Hidden Layer = 2, Epoch = 50) | 2.79272 | 4.63930 | 3.13685 | 3.34318 |
| Neural Network (Hidden Layer = 2, Epoch = 100) | 5.69101 | 7.50384 | 7.56144 | 8.79737 |
| Neural Network (Hidden Layer = 2, Epoch = 150) | 7.02918 | 7.89421 | 8.65984 | 11.27722 |
| Neural Network (Hidden Layer = 3, Epoch = 50) | 3.16504 | 3.10706 | 6.45697 | 3.50108 |
| Neural Network (Hidden Layer = 3, Epoch = 100) | 9.25663 | 5.51506 | 6.18105 | 6.32867 |
| Neural Network (Hidden Layer = 3, Epoch = 150) | 7.45568 | 8.04847 | 12.64496 | 10.19623 |

**Table 16**. Testing time analysis of different algorithms for PID dataset using k-fold cross validation technique.

| Algorithm | Avg. testing time per fold (in second) | | | |
|---|---|---|---|---|
| | 2-Fold | 3-Fold | 5-Fold | 7-Fold |
| MDEM (Mean) | 0.00579 | 0.00354 | 0.00371 | 0.00152 |
| MDEM (Variance) | 0.00869 | 0.00593 | 0.00367 | 0.00267 |
| MDEM (3rd Central Moment) | 0.01191 | 0.00826 | 0.00508 | 0.00354 |
| MDEM (4th Central Moment) | 0.01604 | 0.01063 | 0.00639 | 0.00464 |
| Logistic Regression (LR) | 0.00010 | 0.00010 | 0.00014 | 0.00009 |
| Random Foresh (RF) | 0.00389 | 0.00324 | 0.00515 | 0.00259 |
| Support Vector Machine (SVM) | 0.00097 | 0.00084 | 0.00104 | 0.00053 |
| K Nearest Neighbor (KNN) (n = 3) | 0.00792 | 0.00543 | 0.00731 | 0.00277 |
| K Nearest Neighbor (KNN) (n = 5) | 0.00774 | 0.00536 | 0.00373 | 0.00289 |
| K Nearest Neighbor (KNN) (n = 7) | 0.00791 | 0.00533 | 0.00367 | 0.00730 |
| Neural Network (Hidden Layer = 1, Epoch = 50) | 0.07778 | 0.13250 | 0.07375 | 0.07332 |
| Neural Network (Hidden Layer = 1, Epoch = 100) | 0.07743 | 0.07576 | 0.07357 | 0.12097 |
| Neural Network (Hidden Layer = 1, Epoch = 150) | 0.07589 | 0.07722 | 0.07339 | 0.07369 |
| Neural Network (Hidden Layer = 2, Epoch = 50) | 0.08351 | 0.13375 | 0.07765 | 0.08012 |
| Neural Network (Hidden Layer = 2, Epoch = 100) | 0.08715 | 0.08064 | 0.10729 | 0.09882 |
| Neural Network (Hidden Layer = 2, Epoch = 150) | 0.08379 | 0.08138 | 0.08023 | 0.10519 |
| Neural Network (Hidden Layer = 3, Epoch = 50) | 0.09978 | 0.08951 | 0.14719 | 0.08656 |
| Neural Network (Hidden Layer = 3, Epoch = 100) | 0.15854 | 0.08722 | 0.08827 | 0.08676 |
| Neural Network (Hidden Layer = 3, Epoch = 150) | 0.09376 | 0.09016 | 0.11706 | 0.10369 |

handle massive amount of data e.g., newsfeed generation in social media, email filtering, text classification and categorization, support ticket routing, sentiment analysis, different variants of our proposed MDEM algorithms can come up as a suitable choice with a far better running time and a slightly less than optimal performance.

**Table 17.** **Training time analysis of different algorithms for Wisconsin breast cancer dataset using k-fold cross validation technique.**

| Algorithm | Avg. training time per fold (in second) | | | |
|---|---|---|---|---|
| | 2-Fold | 3-Fold | 5-Fold | 7-Fold |
| MDEM (Mean) | 0.00217 | 0.00259 | 0.00394 | 0.00350 |
| MDEM (Variance) | 0.00441 | 0.00600 | 0.00724 | 0.00847 |
| MDEM (3rd Central Moment) | 0.00706 | 0.00901 | 0.01084 | 0.01178 |
| MDEM (4th Central Moment) | 0.00920 | 0.01209 | 0.01487 | 0.02542 |
| Logistic Regression (LR) | 0.00426 | 0.00363 | 0.00821 | 0.00691 |
| Random Foresh (RF) | 0.08880 | 0.10133 | 0.14573 | 0.11799 |
| Support Vector Machine (SVM) | 0.00085 | 0.00089 | 0.00149 | 0.00120 |
| K Nearest Neighbor (KNN) (n = 3) | 0.00027 | 0.00027 | 0.00030 | 0.00030 |
| K Nearest Neighbor (KNN) (n = 5) | 0.00028 | 0.00028 | 0.00029 | 0.00030 |
| K Nearest Neighbor (KNN) (n = 7) | 0.00028 | 0.00028 | 0.00037 | 0.00029 |
| Neural Network (Hidden Layer = 1, Epoch = 50) | 2.65601 | 2.75188 | 4.38526 | 2.99784 |
| Neural Network (Hidden Layer = 1, Epoch = 100) | 4.71574 | 5.10924 | 5.31371 | 5.39365 |
| Neural Network (Hidden Layer = 1, Epoch = 150) | 6.85594 | 12.29132 | 11.04939 | 7.71052 |
| Neural Network (Hidden Layer = 2, Epoch = 50) | 2.70036 | 2.85225 | 3.06369 | 4.33063 |
| Neural Network (Hidden Layer = 2, Epoch = 100) | 6.73296 | 5.20730 | 5.45749 | 5.56384 |
| Neural Network (Hidden Layer = 2, Epoch = 150) | 7.39526 | 7.61450 | 7.91835 | 7.94657 |
| Neural Network (Hidden Layer = 3, Epoch = 50) | 2.92692 | 3.09993 | 4.27838 | 3.13910 |
| Neural Network (Hidden Layer = 3, Epoch = 100) | 4.98138 | 5.49342 | 5.58660 | 7.26626 |
| Neural Network (Hidden Layer = 3, Epoch = 150) | 7.17968 | 7.86225 | 8.02966 | 8.23422 |

**Table 18.** **Testing time analysis of different algorithms for Wisconsin breast cancer dataset using k-fold cross validation technique.**

| Algorithm | Avg. testing time per fold (in second) | | | |
|---|---|---|---|---|
| | 2-Fold | 3-Fold | 5-Fold | 7-Fold |
| MDEM (Mean) | 0.00930 | 0.00621 | 0.00526 | 0.00276 |
| MDEM (Variance) | 0.01962 | 0.01298 | 0.00816 | 0.00641 |
| MDEM (3rd Central Moment) | 0.02981 | 0.01968 | 0.01167 | 0.00849 |
| MDEM (4th Central Moment) | 0.04015 | 0.02664 | 0.01678 | 0.01700 |
| Logistic Regression (LR) | 0.00013 | 0.00010 | 0.00015 | 0.00011 |
| Random Foresh (RF) | 0.00241 | 0.00226 | 0.00378 | 0.00212 |
| Support Vector Machine (SVM) | 0.00028 | 0.00023 | 0.00027 | 0.00018 |
| K Nearest Neighbor (KNN) (n = 3) | 0.02378 | 0.01668 | 0.01995 | 0.01697 |
| K Nearest Neighbor (KNN) (n = 5) | 0.03063 | 0.01862 | 0.01826 | 0.01578 |
| K Nearest Neighbor (KNN) (n = 7) | 0.03053 | 0.01509 | 0.02185 | 0.02023 |
| Neural Network (Hidden Layer = 1, Epoch = 50) | 0.07349 | 0.07917 | 0.10993 | 0.07281 |
| Neural Network (Hidden Layer = 1, Epoch = 100) | 0.07883 | 0.07481 | 0.07180 | 0.07573 |
| Neural Network (Hidden Layer = 1, Epoch = 150) | 0.07816 | 0.11610 | 0.09967 | 0.06970 |
| Neural Network (Hidden Layer = 2, Epoch = 50) | 0.08087 | 0.08200 | 0.07873 | 0.10247 |
| Neural Network (Hidden Layer = 2, Epoch = 100) | 0.08764 | 0.08066 | 0.07889 | 0.07972 |
| Neural Network (Hidden Layer = 2, Epoch = 150) | 0.16970 | 0.07814 | 0.08098 | 0.07808 |
| Neural Network (Hidden Layer = 3, Epoch = 50) | 0.09919 | 0.08990 | 0.11623 | 0.08566 |
| Neural Network (Hidden Layer = 3, Epoch = 100) | 0.09112 | 0.08646 | 0.08559 | 0.11566 |
| Neural Network (Hidden Layer = 3, Epoch = 150) | 0.09521 | 0.08542 | 0.08595 | 0.08576 |

## Acknowledgments

The author is greatful to the editor and reviewers of the journal for their valuable suggestions.

## Author contributions

**Conceptualization:** Ahmed Mehedi Nizam.

**Data curation:** Ahmed Mehedi Nizam.

**Formal analysis:** Ahmed Mehedi Nizam.

**Investigation:** Ahmed Mehedi Nizam.

**Methodology:** Ahmed Mehedi Nizam.

**Resources:** Ahmed Mehedi Nizam.

**Software:** Ahmed Mehedi Nizam.

**Validation:** Ahmed Mehedi Nizam.

**Visualization:** Ahmed Mehedi Nizam.

**Writing – original draft:** Ahmed Mehedi Nizam.

**Writing – review & editing:** Ahmed Mehedi Nizam.

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
