## [Decision Letter · Decision Letter 0]

30 Jun 2025

PONE-D-25-26429Minimum Displacement in Existing Moment (MDEM)- A new supervised learning algorithm by incrementally constructing the moments of the underlying classesPLOS ONE

Dear Dr. Nizam,

Thank you for submitting your manuscript to PLOS ONE. After careful consideration, we feel that it has merit but does not fully meet PLOS ONE’s publication criteria as it currently stands. Therefore, we invite you to submit a revised version of the manuscript that addresses the points raised during the review process.

We look forward to receiving your revised manuscript.

Kind regards,

Zeheng Wang

Academic Editor

PLOS ONE

Journal Requirements:

Reviewers' comments:

Reviewer's Responses to Questions

**Comments to the Author**

1. Is the manuscript technically sound, and do the data support the conclusions?

Reviewer #1: Partly

Reviewer #2: Yes

Reviewer #3: Yes

Reviewer #4: Yes

2. Has the statistical analysis been performed appropriately and rigorously?

Reviewer #1: Yes

Reviewer #2: Yes

Reviewer #3: Yes

Reviewer #4: No

3. Have the authors made all data underlying the findings in their manuscript fully available?

Reviewer #1: Yes

Reviewer #2: Yes

Reviewer #3: Yes

Reviewer #4: Yes

4. Is the manuscript presented in an intelligible fashion and written in standard English?

Reviewer #1: Yes

Reviewer #2: Yes

Reviewer #3: Yes

Reviewer #4: Yes

5. Review Comments to the Author

Reviewer #1: Review Comments

This paper proposes a supervised learning algorithm based on the existing Moment Displacement Minimum (MDEM), whose core is to classify test data points into the category that minimizes the n-th order central moment displacement of the corresponding category. The algorithm dynamically updates the moments through incremental calculation, enabling the model to have evolutionary capabilities. The study uses the Pima Indians Diabetes Dataset and compares the algorithm with advanced algorithms such as neural networks and support vector machines through k-fold cross-validation and stratified k-fold cross-validation. The results show that the low-order moment MDEM algorithm performs comparably to K-nearest neighbors (KNN) with better test time complexity, providing a new option for data classification problems. However, there are still several issues as follows:

I. Insufficient clarity in expressing core concepts

1. The paper proposes classifying data points into the category with the minimum moment displacement, but it lacks an intuitive explanation of the physical meaning of "moment displacement" and why this index can effectively characterize the correlation between data points and categories. For example, it only states that Euclidean distance is calculated and weighted by the category cardinality, but does not clarify the theoretical basis of the weighting strategy (e.g., why cardinality balances the "attractiveness" of categories), making it difficult for readers to understand the rationality of the decision-making mechanism.

2. The algorithm discusses the applications of the mean (first-order raw moment) and second- to fourth-order central moments, but does not explain why these orders are chosen. Additionally, whether increasing the order will continuously improve classification performance and its boundary conditions are not clearly stated.

II. Limitations in time complexity analysis

1. The paper only theoretically deduces that the training time complexity is O(P) and the testing time is constant, but does not consider the actual overhead of calculating power sums for high-dimensional data (when d is large) or high-order moments (when n is large). For example, when n=4, each data point needs to calculate the 4th power four times. When n and d are large, the time complexity may deviate from the theoretical expectation, and the discussion of boundary conditions is lacking.

2. When comparing with the O(nd) testing complexity of KNN, the paper does not mention the moment calculation cost of MDEM in processing high-dimensional data (e.g., Euclidean distance calculation involves N-dimensional vectors), which may lead to an incomplete understanding of the algorithm's efficiency advantages among readers.

III. Inadequacies in experimental design

1. It only verifies the performance of the mean, variance (second-order central moment), third-order, and fourth-order central moments, but does not set up control groups (such as combinations of mixed-order moments), making it impossible to prove the optimality of a single-order moment or explore the correlation between the order and data features (e.g., whether certain features are more suitable for high-order moments).

2. The paper emphasizes that the model can be updated incrementally, but the experiment does not compare the performance differences between "dynamic update" and "static model" (i.e., not updating moments after training), making it impossible to prove the practical value of this innovation.

3. The comparison metrics only use a single accuracy index, which is difficult to comprehensively measure the model's performance in imbalanced samples. It is recommended to add common machine learning metrics such as F1-score, Precision, Recall, and MCC to comprehensively evaluate the algorithm's performance under multiple metrics.

4. All experiments only use one dataset (PID). The PID dataset used in the experiment has fewer feature dimensions (8 dimensions) and a relatively sufficient sample size (768 samples), making it difficult to prove the algorithm's wide applicability. For example: How does the algorithm perform when facing high-dimensional data? Additionally, how does the algorithm perform on small-sample datasets? Adding these comparisons can further corroborate the algorithm's superiority.

Reviewer #2: In this paper, the author proposes a supervised learning algorithm based upon Minimum Displacement in Existing Moment (MDEM). By some nu- merical experiments, the author evaluates the performances of MDEM in mean, variance, 3th and 4th central moment and obtain accuracy scores 73.43. By the comparison, the author asserts that the MDEM in mean runs better than KNN with 3, 5, 7 neighbors and 8 (eight) of the NN models under consideration in 2-fold cross validation technique, and that the best algorithm under 2-fold cross validation should be SVM, which has a run time accuracy of 76.63.

The manuscript presents a novel supervised learning algorithm called Minimum Displacement in Existing Moment (MDEM) that incrementally constructs class moments. While the concept is something new and well-presented, several as- pects require clarification and improvement before publication.

For more detail, please see the attached review report.

Reviewer #3: Please find detailed comments in the attached file.

This manuscript presents a novel supervised learning algorithm for classi cation called Mini-

mum Displacement in Existing Moment (MDEM). The author claims that the performances

of MDEM involving lower order moments are comparable to those of existing state-of-the-art

classi cation methods.

The algorithm the author proposes can be seen as a variation of the nearest centroid

classi er (or Rocchio algorithm). Instead of calculating the distance between the test data

point to the mean of each class, the proposed algorithm calculates the displacement between

the original mean of each class and the mean after assigning a test data point to each class,

and then pick the class with the least displacement to actually assign the test data point.

Note that the statistic mean can be replaced by other statistics as the author has stated.

The manuscript is well-structured and provides a clear explanation of the algorithm, and

experimental results. However, there are aspects that require clari cation and improvement.

My main concern is the lack of theoretical justi cation for the algorithm, particularly re-

garding the technique of multiplication by the cardinality of the class when calculating the

temporary mean (or other statistics), e.g., line 10 in Algorithm 2.

Overall, the manuscript is a good contribution to the eld of machine learning and

provides a new perspective on classi cation algorithms. I recommend publication and that

the author can address the following major and minor comments.

Reviewer #4: This paper presents a novel and interpretable classification approach by introducing moment-based class inclusion, offering a fresh perspective in supervised learning. MDEM is highly efficient, enabling constant-time classification ideal for real-time or streaming applications, and supports incremental learning by updating class statistics without retraining. The algorithm is transparently described with clear mathematical formulation and full pseudocode, and its performance is thoroughly benchmarked against a diverse set of standard models using multiple validation strategies.

While it doesn't outperform top-tier classifiers like SVMs or deep neural networks, its simplicity and speed make it a valuable tool for constrained environments or streaming applications. Broader testing and deeper theoretical analysis would strengthen the proposal's impact.

Three such suggestions are:

1. All results are from the Pima Indian Diabetes dataset. This limits generalizability. Benchmarking on diverse datasets to stress-test MDEM might be helpful.

2. MDEM’s design lacks the flexibility or capacity of nonlinear models like SVM (with kernels) or deep NNs. This inherently caps its accuracy, especially in more complex domains. Would it be possible to combine MDEM with lightweight NNs or tree-based models to handle non-linearity? Kernel tricks may also be able to extend MDEM to non-linear decision boundaries.

3. The intuition behind using minimal moment displacement is compelling but not theoretically grounded. Formal proofs of convergence, error bounds, or probabilistic guarantees are missing.

6. PLOS authors have the option to publish the peer review history of their article (what does this mean?). If published, this will include your full peer review and any attached files.

Reviewer #1: No

Reviewer #2: **Yes: **Changqing XU

Reviewer #3: No

Reviewer #4: No

---

## [Author Response · Author response to Decision Letter 1]

8 Aug 2025

Response to reviewers

Ahmed Mehedi Nizam August 8, 2025

1 Reviewer-1

1.1 Comment-I.1

Reviewer: 1

I) Insufficient clarity in expressing core concepts

• The paper proposes classifying data points into the category with the min- imum moment displacement, but it lacks an intuitive explanation of the physical meaning of ”moment displacement” and why this index can effec- tively characterize the correlation between data points and categories. For example, it only states that Euclidean distance is calculated and weighted by the category cardinality, but does not clarify the theoretical basis of the weighting strategy (e.g., why cardinality balances the ”attractiveness” of categories), making it difficult for readers to understand the rationality of the decision-making mechanism.

Author:

We will attain the reviewer’s concern in the following two ways:

1. Now, we have provided a more detailed intuitive explanation of the physi- cal meaning of ”moment displacement” in section 2 of the revised manuscript. To be precise, in section 2, now we have explicitly mentioned that the 3rd and 4th central moment are non-normalized skewness and kurtosis of the distribution. There were no physical interpretations of the 3rd and 4th order moments in the previous version of the manuscript. Now, we have added it. Moreover, in the previous as well as the current version of the manuscript, we have mentioned that the first raw moment and second central moment as used in the analysis are mean and variance of the dis- tribution respectively. So, whenever we say ”moment displacement”, we seek to apply some sort of ”Minimum Perturbation Criteria”, i.e., inser- tion of new data point into the class that is least perturbed due to such inclusion in terms of its existing mean, variance, skewness and kurtosis,

(which one is applicable). It is to be mentioned in this regard that ”Min- imum Perturbation Criterion” is widely used in machine learning as well as in other disciplines.

2. To justify the weighting strategy, we have added a whole new section titled ”MDEM as a size-weighted minimum perturbation classifier”, which establishes MDEM algorithm as a size-weighted minimum perturbation classifier. To justify the weighting strategy of the displacement in moments by the respective class cardinality, here we have given a formal proof of the fact that the displacement in the n-th central moment and displacement in 1st raw moment (mean) of a class due to inclusion of a new instance, are inversely proportional to the respective class cardinality. So, to obtain an unbiased measure of displacement metric, we need to multiply it by the respective class cardinality. The formal proof is given in section 5 of the revised manuscript.

1.2 Comment-I.2

Reviewer: 1

The algorithm discusses the applications of the mean (first-order raw moment) and second- to fourth-order central moments, but does not explain why these orders are chosen. Additionally, whether increasing the order will continuously improve classification performance and its boundary conditions are not clearly stated.

Author:

The first raw moment as well as 2nd, 3rd and 4th central moments are mean, variance, skewness and kurtosis of the distribution, each of which have specific statistical interpretations. Moments, higher in order than the 4th order mo- ments, are known as hyper-skewness (odd ones) and hyper-kurtosis (even one). These higher order moments are rarely used in statistical analysis and only capture fine-details of skewness and kurtosis respectively. We have added the following paragraph in the ’Methodology’ section of our manuscript to address the issue:

”Here, we consider only first raw moment as well as 2nd, 3rd and 4th order central moments as they capture some specific attributes of the underlying dis- tribution, namely, mean, variance, skewness and kurtosis respectively. It is to be noted in this regard that moments, higher in order than the 4th order moments, are known as hyper-skewness (odd ones) and hyper-kurtosis (even one) that are rarely used in statistical analysis and only capture fine details of skewness and kurtosis respectively.”

Moreover, whether the performance of MDEM algorithms increases or decreases with the order of moments, depends on the underlying dataset used in the analysis. If the dataset favors lower order moments over higher order ones, then using lower order moments delivers better performance and, in our cases, we have reported exactly this incident. However, there may be other datasets that

prefer homogeneity in skewness and kurtosis to that in mean and variance and, for those datasets, MDEM involving higher order moments may perform better

- We have added these reasonings in the ’Discussion’ subsection under ’Results’ section of our revised manuscript.

1.3 Comment II

Reviewer: 1

Limitations in time complexity analysis

1. The paper only theoretically deduces that the training time complexity is O(P) and the testing time is constant, but does not consider the actual overhead of calculating power sums for high-dimensional data (when d is large) or high-order moments (when n is large). For example, when n=4, each data point needs to calculate the 4th power four times. When n and d are large, the time complexity may deviate from the theoretical expectation, and the discussion of boundary conditions is lacking.

2. When comparing with the O(nd) testing complexity of KNN, the paper does not mention the moment calculation cost of MDEM in processing high-dimensional data (e.g., Euclidean distance calculation involves N- dimensional vectors), which may lead to an incomplete understanding of the algorithm’s efficiency advantages among readers.

Author:

1. In section: 4.1 (Time complexity of MDEM in mean) of our manuscript, we have mentioned that the initial training phase time complexity of MDEM in mean is actually O(N ×P ), where P is the number of training rows and N is the number of attributes. Then, we discuss the issue for a specific problem. For a specific problem, the number of attributes, N , does not change. So, as we have discussed in section: 4.1., the time complexity reduces to simply O(P ) for a single problem under consideration.

Moreover, in section of 4.1 (Time complexity of MDEM in mean), we have also discussed that the training phase time complexity of MDEM in mean is actually O(M × N ), where M is the number of classes and N is the number of attributes. M and N are fixed for a specific problem. So, if we confine ourselves to a specific problem, then the testing phase time complexity is constant. But, in general the time complexity is still O(M × N ) as a whole as we discussed in section: 4.1.

Additionally, in section: 4.2 (Time complexity of MDEM in n-th central moment), we mention that the initial training phase time complexity of MDEM in n-th central moment is actually O(n × N × P ), where n is the number of moments, N is the number of attributes and P is the number of training instances. For a specific problem the value of N is fixed and for

a specific implementation of MDEM algorithm, n is fixed, which reduces the time complexity to O(P ). However, as we have discussed in section: 4.2., the original time complexity remains at O(n × N × P ). Moreover, as we have discussed in section: 4.2., the testing phase time complexity of the algorithm is O(n × N × M ), where n is the number of moment, N is the number of attributes and M is the number of classes. For a specific problem, the value of N and M are fixed and for a specific implementation of MDEM algorithm, n is fixed, which reduces the time complexity to a constant, i.e., it does not depend upon any variable and is therefore fixed for every new testing instance. However, as we have discussed in section: 4.2, the actual testing phase time complexity stays at O(n × N × M ).

2. In section: 4.1, we have discussed that the testing phase time complexity of MDEM in mean is actually O(N × M ), where N is the number of attributes and M is the number of classes. Moreover, in section: 4.2., we have described the testing phase time complexity of MDEM in n-th central moment is O(n × N × M ), where n is the number of moment, N is the number of attributes and M is the number of classes. We think, explanations provided in section: 4.1 and 4.2, should resolve any such ambiguity.

1.4 Comment-III

Reviewer: 1 III. Inadequacies in experimental design

1. It only verifies the performance of the mean, variance (second-order central moment), third-order, and fourth-order central moments, but does not set up control groups (such as combinations of mixed-order moments), making it impossible to prove the optimality of a single-order moment or explore the correlation between the order and data features (e.g., whether certain features are more suitable for high-order moments).

2. The paper emphasizes that the model can be updated incrementally, but the experiment does not compare the performance differences between ”dynamic update” and ”static model” (i.e., not updating moments after training), making it impossible to prove the practical value of this inno- vation.

3. The comparison metrics only use a single accuracy index, which is difficult to comprehensively measure the model’s performance in imbalanced sam- ples. It is recommended to add common machine learning metrics such as F1-score, Precision, Recall, and MCC to comprehensively evaluate the algorithm’s performance under multiple metrics.

4. All experiments only use one dataset (PID). The PID dataset used in the experiment has fewer feature dimensions (8 dimensions) and a relatively sufficient sample size (768 samples), making it difficult to prove the algo- rithm’s wide applicability. For example: How does the algorithm perform

when facing high-dimensional data? Additionally, how does the algorithm perform on small-sample datasets? Adding these comparisons can further corroborate the algorithm’s superiority.

Author:

1. The reviewer has rightly pointed out that we do not go for checking opti- mality of single order moments. However, as we have mentioned in our re- vised manuscript that MDEM is designed for speed and not for optimality. It performs with greater speed delivering only sub-optimal performances. In our revised manuscript, we have added a subsection titled ’Discussion’ under result section in this regard, which reads as follows.

”MDEM algorithms are better suited in numerous situations, where speed in running time is more important than cutting-edge accuracy. For exam- ple, MDEM algorithms can be practically deployed to different text clas- sification scenarios, e.g., classifying documents according to search query, email filtering (spam vs non-spam and also categorization and labelling of emails), categorization of news articles (sports, politics, international etc.), populating newsfeeds for users in social media, sentiment analysis (classifying reviews as positive, negative and neutral), support ticket rout- ing (assign incoming tickets to the right department), legal and medical documents tagging (tag documents based on predefined categories) and the alike. These are some of the applications, where speed is preferred to optimality as the data here are massive in quantity and consequences of possible misclassification are relatively less dangerous.”

In addition, we have also discussed that the performance of MDEM algo- rithms may vary depending upon the underlying dataset. If the dataset prefers mean and variances over higher order moments like skewness and kurtosis, then MDEM involving lower order moments will perform well. However, if the dataset prefers skewness and kurtosis over lower order moments, then MDEM with higher order moments will perform well. We have discussed the issue in the 1st paragraph newly added ”Discussion” subsection in ”Results” section of our revised manuscript, that reads as follows.

”However, which version of MDEM algorithm will perform better than the others depends mostly on the underlying dataset used in the analysis. If the dataset favors lower order moments like mean and variance over higher order ones like skewness and kurtosis, then using lower order moments delivers better performance and, in our cases, we have reported exactly the same incident. However, there may be other datasets that prefer homogeneity in skewness and kurtosis to that in mean and variance and, for those datasets, MDEMs involving higher order moments may perform better.”

2. As the model can effectively train itself in constant time with a simple assignment of temporary moment to the new moment, we think, we should go for the ’dynamic one’ as it learns continuously without incurring much overhead.

3. To address the reviewer’s concern, we have now added F1-score, Precision, Recall, and MCC in our analysis of the performance of MDEM algorithms.

4. We have now extended our analysis using Wisconsin breast cancer dataset, which has 30 features and 569 instances. We think this will address the reviewer’s concerns involving high-dimensional and relatively small sample datasets. Results for Wisconsin breast cancer dataset are even better than that of PID dataset as we have discussed the issue in the revised manuscript.

2 Reviewer-2: C Xu

2.1 Comment-1

Reviewer-2: C Xu: Methodological Clarifications

1. The manuscript would benefit from clearer pseudocode formatting in Algo- rithms 1-4. Consider using standard algorithmic packages like algorithm2e for better readability.

2. The moment calculation formulas (Equations 1-4) are correct but could be more clearly connected to the algorithmic implementation. A small numerical example showing moment calculation would be helpful.

Author

1. As per reviewer’s suggestion, we have now reformatted algorithm: 1-4 using algorithm2e package.

2. Following reviewer’s suggestion, a numerical example of moment calcula- tion has been embedded at the beginning of section: 3.2 (MDEM in n-th central moment).

2.2 Comment-2

Reviewer-2: C Xu: Empirical Evaluation

1. The performance range of 83.19%-95.82% of the best algorithm is notable, but the paper should discuss why higher-order moments (3rd, 4th) under- perform compared to mean and variance.

2. The comparison with neural networks mentions performance differences based on layers/nodes, but doesn’t explain why MDEM outperforms smaller NNs while being outperformed by larger ones. This deserves deeper anal- ysis.

3. The results tables (Tables 2-3) would be more impactful with statistical significance indicators (e.g., p-values or confidence intervals).

Author:

1. Following the reviewer’s suggestion, we have added a ’Discussion’ subsec- tion under ’Results’ section in our manuscript that discusses the issue. It reads as follows.

”From the above discussion, it can be seen that the MDEM algorithm performs within the [83.19% − 95.82%] bound of the best algorithm un- der consideration for PID dataset. Moreover, for Wisconsin breast can- cer dataset, its performance seems to swing between [88.85% − 96.41%]. Moreover, for both the datasets, MDEMs involving lower order moments perform remarkably well as compared to MDEMs involving higher order moments. In fact, the lower bounds of the afore-mentioned accuracy scores are attributed to MDEMs involving 3rd and 4th order central moments, while the upper bounds are practically due to MDEMs in mean. Also, as we can be seen from Table: [5], [8], [11] and [14], MDEMs in mean and variance have consistently obtained better F1 score and MCC as compared to MDEMs in skewness and kurtosis. However, which version of MDEM algorithm will perform better than the others depends mostly on the un- derlying dataset used in the analysis. If the dataset favors lower order moments like mean and variance over higher order ones like skewness and kurtosis, then using lower order moments delivers better performance and, in our cases, we have reported exactly the same incident. However, there may be other datasets that prefer homogeneity in skewness and kurtosis to that in mean and variance and, for those datasets, MDEMs involving higher order moments may perform better.”

2. The reviewer has rightly pointed out that MDEM outperforms smaller NNs, while being outperformed by larger ones in case of PID dataset. T

---

## [Decision Letter · Decision Letter 1]

24 Aug 2025

PONE-D-25-26429R1Minimum Displacement in Existing Moment (MDEM)- A new supervised learning algorithm by incrementally constructing the moments of the underlying classesPLOS ONE

Dear Dr. Nizam,

Thank you for submitting your manuscript to PLOS ONE. After careful consideration, we feel that it has merit but does not fully meet PLOS ONE’s publication criteria as it currently stands. Therefore, we invite you to submit a revised version of the manuscript that addresses the points raised during the review process.

We look forward to receiving your revised manuscript.

Kind regards,

Zeheng Wang

Academic Editor

PLOS ONE

Journal Requirements:

Reviewers' comments:

Reviewer's Responses to Questions

**Comments to the Author**

1. If the authors have adequately addressed your comments raised in a previous round of review and you feel that this manuscript is now acceptable for publication, you may indicate that here to bypass the “Comments to the Author” section, enter your conflict of interest statement in the “Confidential to Editor” section, and submit your "Accept" recommendation.

Reviewer #2: All comments have been addressed

Reviewer #3: All comments have been addressed

Reviewer #4: All comments have been addressed

2. Is the manuscript technically sound, and do the data support the conclusions?

Reviewer #2: Yes

Reviewer #3: Yes

Reviewer #4: Yes

3. Has the statistical analysis been performed appropriately and rigorously?

Reviewer #2: Yes

Reviewer #3: Yes

Reviewer #4: Yes

4. Have the authors made all data underlying the findings in their manuscript fully available?

Reviewer #2: Yes

Reviewer #3: Yes

Reviewer #4: Yes

5. Is the manuscript presented in an intelligible fashion and written in standard English?

Reviewer #2: Yes

Reviewer #3: Yes

Reviewer #4: Yes

6. Review Comments to the Author

Reviewer #2: It would be better if a short discussion on convergence/error bounds could be added and runtime analysis on higher- dimensional data could be supplied (both are optional).

Reviewer #3: The author has fully justified my previous comments and concerns. I would recommend for publication.

Note: I would like to keep my comments anonymous to public. Thanks!

Reviewer #4: The author has thoroughly addressed all of my concerns, and I have no further comments. This manuscript is acceptable for publication.

7. PLOS authors have the option to publish the peer review history of their article (what does this mean?). If published, this will include your full peer review and any attached files.

Reviewer #2: **Yes: **Changqing Xu

Reviewer #3: No

Reviewer #4: No

---

## [Author Response · Author response to Decision Letter 2]

17 Oct 2025

Reviewer 2: It would be better if a short discussion on convergence/error bounds could be added and runtime analysis on higher- dimensional data could be supplied (both are optional).

Author:

1. As per reviewer's suggestion, we have now described a sufficient condition for optimality under minimum perturbation criteria. This is included into section 6 of our revised manuscript.

2. As per reviewer's suggestion, we have now added a new subsection dedicating to the run time analysis of our proposed algorithm as well as all other algorithms under consideration. This is included into subsection 10.3 and Table [15]-[18] in our revised manuscript.

---

## [Editor Report · Decision Letter 2]

2 Nov 2025

Minimum Displacement in Existing Moment (MDEM)- A new supervised learning algorithm by incrementally constructing the moments of the underlying classes

PONE-D-25-26429R2

Dear Dr. Nizam,

We’re pleased to inform you that your manuscript has been judged scientifically suitable for publication and will be formally accepted for publication once it meets all outstanding technical requirements.

Kind regards,

Zeheng Wang

Academic Editor

PLOS ONE
---

## [Editor Report · Acceptance letter]

PONE-D-25-26429R2

PLOS ONE

Dear Dr. Nizam,

I'm pleased to inform you that your manuscript has been deemed suitable for publication in PLOS ONE. Congratulations! Your manuscript is now being handed over to our production team.

Kind regards,

on behalf of

Dr. Zeheng Wang

Academic Editor

PLOS ONE